# CONTRACTION AND ALIENATION: TOWARDS THEORETICAL UNDERSTANDING NON-CONTRASTIVE LEARNING WITH NEIGHBOR-AVERAGING DYNAMICS

## ABSTRACT

Non-contrastive self-supervised learning (SSL) is a popular paradigm for learning representations by explicitly aligning positive pairs. However, due to specialized implementation details, the underlying working mechanism of non-contrastive SSL remains somewhat mysterious. In this paper, we investigate the implicit bias of non-contrastive learning with a concise framework, namely SimXIR. SimXIR optimizes the online network by alternatively taking the online network of the last round as the target network, without requiring asymmetric tricks and momentum updates. Notably, the expectation minimization inherent to SimXIR can be reformulated as the *neighbor-averaging dynamics*, in which each representation is iteratively replaced with the average representation of its neighbors. Moreover, we introduce the concept of neighbor-connected groups that organize samples through the neighboring paths on data, and assume the input sample space is composed of multiple disjoint neighbor-connected groups. We theoretically prove that the concise dynamics of SimXIR exhibit two intriguing properties: *contraction of neighbor-connected groups* and *alienation between disjoint groups*, which resemble intra-class compactness and inter-class separability in classification and help explain why non-contrastive SSL can prevent collapsed solutions. Inspired by the theoretical results, we propose a novel step for self-supervised pre-training—self-supervised fine-tuning, and leverage SimXIR to further enhance representations of off-the-shelf SSL models. Experimental results demonstrate the effectiveness of SimXIR in improving self-supervised representations, ultimately achieving better performance on downstream classification tasks.

## 1 INTRODUCTION

Learning high-quality representations plays a critical role in various computer vision tasks. Traditional supervised learning heavily relies on extensive labeled data, which can be expensive or unattainable in practical scenarios (Deng et al., 2009a). In recent years, self-supervised representation learning has emerged as a potent alternative, delivering competitive or even superior performance in downstream tasks compared to supervised methods, while without relying on human annotations (Bardes et al., 2022; Tomasev et al., 2022; Xie et al., 2022; Assran et al., 2023; Oquab et al., 2023).

One prevalent paradigm in self-supervised learning (SSL) within computer vision involves generating multiple views of the same image through diverse hand-crafted data augmentations. These views are then optimized to agree under different conditions. Contrastive learning, a popular line of SSL, aims to bring the representations of different views of the same sample (positive pairs) closer together while pushing apart the representations of views from different samples (negative pairs) (Ye et al., 2019; Chen et al., 2020; He et al., 2020; Tao et al., 2022). Contrasting negative pairs serves the crucial role of preventing trivial solutions for contrastive learning, where all data representations collapse into a single vector. Another branch of SSL takes a non-contrastive approach, which focuses solely on aligning positive pairs to capture the invariance of transformations or distortions (Grill et al., 2020; Chen and He, 2021; Caron et al., 2021; Zbontar et al., 2021). To avoid collapsed solutions, state-of-the-art non-contrastive SSL methods rely on specialized network architectures, including asymmetric structures between online and target networks (using projection and prediction heads)

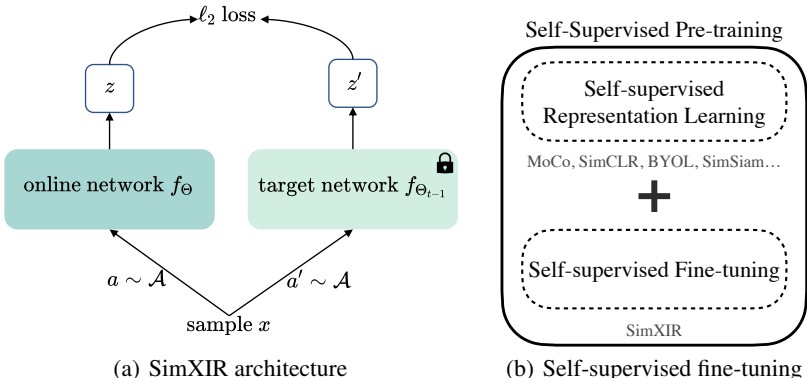

(a) SimXIR architecture

(b) Self-supervised fine-tuning

Figure 1: Illustrations of SimXIR and self-supervised fine-tuning. (a) **SimXIR architecture**. Two augmented views of one sample are processed by the online network $f_\Theta$ and the target network $f_{\Theta_{t-1}}$ frozen (*i.e.*, not be updated) as the online network from the last round. The model minimizes $\ell_2$ loss between both sides. It uses neither negative pairs nor additional MLP layers as projector and predictor. (b) **New pre-training paradigm**. It includes the standard self-supervised representation learning and the proposed SimXIR for self-supervised fine-tuning to achieve boosted representations.

(Grill et al., 2020; Caron et al., 2021), momentum updates for the target network (Tarvainen and Valpola, 2017; He et al., 2020), and Siamese network architectures (Chen and He, 2021).

While non-contrastive SSL achieves remarkable performance without negative pairs, the underlying mechanics remains somewhat mysterious. Several studies in the literature have attempted to provide theoretical investigations on non-contrastive SSL using simplified models (Tian et al., 2021; Wang et al., 2021; Wei et al., 2021; Wen and Li, 2022; Zhang et al., 2022; Zhuo et al., 2023; Xue et al., 2023). For instance, Tian et al. (2021) utilized a two-layer linear model to explain why BYOL (Grill et al., 2020) and SimSiam Chen and He (2021) representations do not collapse to zero. Wang et al. (2021) demonstrated that non-contrastive SSL in linear networks can reduce sample complexity on downstream tasks. Zhuo et al. (2023) hypothesize that asymmetric modules behave as low-pass online spectral filters which create a rank difference in features to alleviate either complete or dimensional feature collapse. Nevertheless, these investigations have primarily focused on the asymmetric designs in non-contrastive SSL and have not delved into the fundamental principles behind the paradigm of only aligning positive pairs. Consequently, a pertinent question arises:

*Is there an implicit bias that allows non-contrastive SSL to avoid collapsed solutions even in the absence of asymmetric designs?*

In this paper, we make the first attempt by introducing a **Sim**ple and e**X**plainable framework for understanding and **I**mproving self-supervised **R**epresentations without any asymmetric designs, namely **SimXIR**. As illustrated in Fig. 1(a), SimXIR optimizes the online network (which only contains the encoder) by alternatively taking the online network of the last round as the target network, without asymmetric designs between two branches. This streamlined framework offers more mathematical opportunities to understand the inherent properties of non-contrastive SSL. We show that SimXIR can be reformulated as the *neighbor-averaging dynamics* which involves a neighborhood-based message interaction like label propagation (Zhu et al., 2003), that is, each representation is iteratively replaced with the average representation in the neighborhood.

One of the key contributions of this paper is the revelation of the implicit bias inherent in SimXIR, specifically the identification of *contraction* and *alienation* properties in non-contrastive SSL. Although SimXIR may seem simplistic at first glance due to the lack of sophisticated mechanisms to prevent collapsed solutions, our analysis surprisingly reveals that it carries an implicit bias against representation collapsing when viewed through the lens of *neighbor-averaging dynamics*. In this context, we introduce the concept of neighbor-connected groups in the data, a perspective of grouping samples based on their connectedness through neighboring paths. Notably, these neighbor-connected groups are defined by intrinsic data properties rather than semantic labels, broadening our perspective beyond traditional notions like "class" and enhancing the theoretical analysis of SSL. Furthermore, we introduce the separation assumption, asserting that the input sample space comprises disjoint neighbor-connected groups, which extends the classical *instance-based separability* concept to a

novel form: *group-based separability*. Building on this definition and assumption, we theoretically establish that the concise dynamics of SimXIR embody two compelling properties:

(i) ***Contraction of neighbor-connected groups***, indicating a reduction in the maximal distance between representations within a group, thereby facilitating the alignment of positive pairs.

(ii) ***Alienation between disjoint groups***, signifying an increase in the minimal distance between two disjoint groups. This property helps prevent all representations from collapsing into a constant value.

In analogy with the concepts of intra-class compactness and inter-class separability in classification tasks, these two properties theoretically support SimXIR's ability to learn discriminative representations. This discovery unveils a novel implicit bias in non-contrastive SSL: *SimXIR implicitly brings representations of samples in the same neighbor-connected group closer together while simultaneously pushing those of disjoint groups further apart, effectively preventing collapsed solutions.*

To validate the effectiveness of SimXIR, we introduce a novel step in self-supervised pre-training, namely **self-supervised fine-tuning**, instead of relying on SimXIR as a standalone framework for training a model from random initialization. To the best of our knowledge, this approach has not been explored in the existing literature. As empirically demonstrated in Section 5, self-supervised fine-tuning in combination with SimXIR exhibits the potential to further enhance representations compared to various off-the-shelf SSL models.

Our main contributions are highlighted as follows:

– We introduce SimXIR, a simple yet theoretically sound framework for understanding and enhancing non-contrastive SSL. This innovative framework dispenses with the need for additional MLP layers, maintaining a concise form through neighbor-averaging dynamics. Additionally, it serves as a valuable self-supervised fine-tuning module to enhance representations across various SSL models.

– We introduce the concept of neighbor-connected groups and conduct a thorough theoretical examination of the role played by SimXIR-induced neighbor-averaging dynamics. Furthermore, we identify two critical properties associated with our framework: the contraction of neighbor-connected groups and the alienation between disjoint groups. This discovery sheds light on a new implicit bias of non-contrastive SSL and aids in understanding why it avoids collapsing to constant solutions.

– We offer comprehensive experiments involving the fine-tuning of off-the-shelf SSL models with SimXIR. The results, which encompass downstream linear classification and kNN classification, demonstrate the effectiveness of SimXIR in elevating self-supervised representations.

## 2 NEIGHBOR-AVERAGING DYNAMICS

**Notation.** First of all, we offer definitions of some notations to facilitate the subsequent formulation. We denote $\mathcal{X}$ as the input space that can be implicitly partitioned into a few groups, and denote $\mathcal{A}$ as the set of transformations composed of various data augmentations. As illustrated in Fig. 2, we define $\mathcal{B}(x) = \{x' : \exists a \in \mathcal{A} \text{ such that } \|x' - a(x)\|_2 \leq r\}$ as the points within distance $r$ from certain data augmentations of a sample $x$ drawn from $\mathcal{X}$; define the neighborhood of $x$ as $\mathcal{N}(x)$, which consists of the set of points whose transformation sets overlap with that of $x$:

$$\mathcal{N}(x) = \{x' : \mathcal{B}(x') \cap \mathcal{B}(x) \neq \emptyset\}. \qquad (2.1)$$

We can further define the $n$-th ($n \geq 2$) order neighborhood of $x$ recursively:

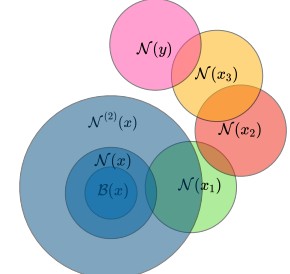

Figure 2: Illustration of some concepts $\mathcal{B}(x)$, $\mathcal{N}(x)$, $\mathcal{N}^{(2)}(x)$, and a neighboring path between $x$ and $y$, where $x_1 \in \mathcal{N}^{(2)}(x)$.

$$\mathcal{N}^{(n)}(x) = \{x' : \mathcal{N}(x') \cap \mathcal{N}^{(n-1)}(x) \neq \emptyset\}, \qquad (2.2)$$

where the 1-st order neighborhood $\mathcal{N}^{(1)}(x) = \mathcal{N}(x)$. Obviously, it can be derived that $x \in \mathcal{B}(x) \subseteq \mathcal{N}^{(1)}(x) \subseteq \cdots \subseteq \mathcal{N}^{(n-1)}(x) \subseteq \mathcal{N}^{(n)}(x)$.

**Non-contrastive Self-supervised Learning.** Different from contrastive approaches (Chen et al., 2020; He et al., 2020) that encourage closer representations of positive pairs than negative pairs, non-contrastive SSL works by enlarging agreement between positive pairs only. Representative

non-contrastive SSL schemes include BYOL (Grill et al., 2020) and SimSiam (Chen and He, 2021), for which a unified SSL objective can be formulated as

$$\mathbb{E}_{x\in\mathcal{X}}\mathbb{E}_{x_1,x_2\in\mathcal{A}(x)}L(f_\Theta(x_1),g_{\Theta'}(x_2)) \approx \mathbb{E}_{x\in\mathcal{X}}\mathbb{E}_{x'\in\mathcal{N}(x)}L\left(f_\Theta(x),g_{\Theta'}(x')\right), \tag{2.3}$$

where $L$ represents the loss function (usually using the $\ell_2$ loss), $f_\Theta$ and $g_{\Theta'}$ denote the online and target networks, respectively. In non-contrastive SSL, the goal is to optimize parameters $\Theta$ of the online network by minimizing the loss above with the target network $g_{\Theta'}$ being frozen with the stop-gradient operation. To prevent collapsed solutions, theoretical and empirical results show that it is necessary to design $f_\Theta$ and $g_{\Theta'}$ with asymmetric structures (Tian et al., 2021).

As shown in Eq. (2.3), for each $x \in \mathcal{X}$, we optimize $\Theta$ by minimizing $\mathbb{E}_{x'\in\mathcal{N}(x)}\|f_\Theta(x) - g_{\Theta'}(x')\|_2^2$. By further deriving this expression, we can obtain an equivalent formulation, which involves minimizing $\|f_\Theta(x) - \mathbb{E}_{x'\in\mathcal{N}(x)}g_{\Theta'}(x')\|_2^2$ to optimize $\Theta$. It is worth noting that, (Grill et al., 2020; Chen and He, 2021) have demonstrated that the conditional risk minimizer can be easily derived as $f_{\Theta^*}(x) = \mathbb{E}_{x'\in\mathcal{N}(x)}g_{\Theta'}(x')$. It indicates that $f_{\Theta^*}(x)$ can be represented as the average of target representations in the neighborhood $\mathcal{N}(x)$.

**Neighbor-Averaging Dynamics.** Although the above form is concise and aesthetically pleasing, some of its underlying properties remain not well understood. This is primarily due to the intractability of neural networks, including the asymmetric structures of the online and target networks (Grill et al., 2020; Caron et al., 2021), the momentum updates of the target network (Tarvainen and Valpola, 2017; He et al., 2020; Chen et al., 2021), and Siamese network (Bromley et al., 1993; Chen and He, 2021). The limitations of the existing non-contrastive SSL motivate us to pursue an even more succinct and explainable framework. To this end, we introduce a simple, explainable and theoretically sound framework (*a.k.a*, SimXIR) to understand and improve self-supervised representations. The main idea is to substitute the target network $g_{\Theta'}$ with the online network of the last round, thus the conditional risk minimizer is reformulated as a series of iterations involving the replacement of representations with their neighborhood-based average:

$$f_{\Theta_t}(x) = \mathbb{E}_{x'\in\mathcal{N}(x)}f_{\Theta_{t-1}}(x'). \tag{2.4}$$

It is worth noting that throughout this paper, for brevity we continue to denote the online network as $f_\Theta$. However, unlike BYOL, SimSiam and Hua et al. (2021), $f_\Theta$ only contains the encoder, without additional MLP layers as the projector and predictor. Despite its apparent simplicity, this replacement plays a crucial role in eliminating asymmetric tricks and providing more mathematical opportunities to comprehend the intrinsic properties of non-contrastive SSL. In the subsequent sections, we will offer a detailed introduction to the proposed framework and its theoretical insights.

**Remark.** Beyond the realm of self-supervised learning, the neighbor-averaging dynamics in Eq. (2.4) can be seen as a neighborhood-based message interaction, akin to affinity propagation (Frey and Dueck, 2007) in clustering and label propagation (Zhu et al., 2003) in semi-supervised learning. For instance, when the representation function $f$ is reduced to the label function, the neighbor-averaging dynamics encompass the iterative process of label propagation. In this context, it can be expressed as $f_t(x) = \sum_{x'} p(x'|x)f_{t-1}(x')$, where $p(x'|x)$ that denotes the probability of $x'$ being in the neighborhood of $x$ can be regarded as the edge weight between two vertices $x'$ and $x$. As shown in Fig. 3, this scheme further boosts the clustering performance, such as k-Means from 89% to 96%.

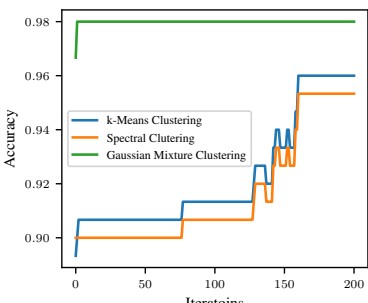

Figure 3: Clustering accuracy of performing neighbor-averaging dynamics (where $\mathcal{N}(x) = \{x' : \|x' - x\| \leq 0.4\}$) after clustering on Iris (Fisher, 1936).

## 3 METHODOLOGY

In this section, we elaborate the proposed method that coincides closely with the dynamics in Eq. (2.4). Our method not only offers a streamlined framework to provide mathematical insights for non-contrastive SSL, but also can serve as a useful self-supervised fine-tuning module under the paradigm of self-supervised pre-training.

The architecture of the proposed SimXIR is illustrated in Fig. 1(a). Two randomly augmented views $a(x)$ and $a'(x)$ from a sample $x$ are taken as input, which are subsequently encoded by the online

network $f_\Theta$ and the target network $f_{\Theta_{t-1}}$ (the frozen online network from the last round) to generate two representations $z$ and $z'$, respectively. The produced two embedding vectors are used to optimize the online network $f_\Theta$ by minimizing the $\ell_2$ loss: $\ell_2(z, z') = \|z - z'\|_2^2$, where $\|\cdot\|_2$ denotes the $\ell_2$-norm. The parameter $\Theta$ is optimized by minimizing the following expected risk:

$$\mathcal{R} = \mathbb{E}_{x \in \mathcal{X}} \mathbb{E}_{a, a' \sim \mathcal{A}} \|f_\Theta(a(x)) - f_{\Theta_{t-1}}(a'(x))\|_2^2. \tag{3.1}$$

**Self-supervised Fine-tuning.** Compared with BYOL (Grill et al., 2020), SimSiam (Chen and He, 2021), and the framework $\mathcal{L}(\Theta) = \mathbb{E}_{x \in \mathcal{X}} \mathbb{E}_{a, a' \sim \mathcal{T}} \|f_\Theta(a(x)) - f_\Theta(a'(x))\|_2^2$ proposed in (Hua et al., 2021), SimXIR distinguishes itself by utilizing $f_{\Theta_{t-1}}$, the frozen online network from the previous round, as the target network for updating the current online network. Notably, it does not necessitate additional MLP layers or surrogate objective functions. As depicted in Fig. 1(b), SimXIR introduces a novel step in self-supervised pre-training, namely self-supervised fine-tuning. Instead of functioning as a standalone framework for training a model from random initialization, SimXIR complements the existing SSL process, leading to enhanced representations, as demonstrated by experimental results in Section 5. The operational flow of SimXIR is detailed in Algorithm 1 in the Appendix.

**A Variant of SimXIR.** By reformulating the expected risk in Eq. (3.1) with respect to the sample $x$ and its neighbors, we can obtain the following expression:

$$\mathcal{R} \approx \mathbb{E}_{x \in \mathcal{X}} \mathbb{E}_{x' \in \mathcal{N}(x)} |f_\Theta(x) - f_{\Theta_{t-1}}(x')|2^2 = \mathcal{R}', \tag{3.2}$$

which indicates that the risk is approximately equal to the expectation of the conditional risk $\mathcal{R}_c(x) = \mathbb{E}_{x' \in \mathcal{N}(x)} |f_\Theta(x) - f_{\Theta_{t-1}}(x')|_2^2$, i.e., $\mathcal{R}' = \mathbb{E}_{x \in \mathcal{X}}[\mathcal{R}_c(x)]$. This formula also gives rise to a variant of SimXIR, which is used for self-supervised fine-tuning and leverages $\mathcal{R}'$ as the objective function to optimize the online network. The empirical version of $\mathcal{R}'$ can be easily implemented by feeding the original sample into the online network and one of its neighbors into the target networks. Specifically, Step 7 in Algorithm 1 can be replaced with $\ell_i \leftarrow |f_\Theta(x_i) - f_{\Theta_{t-1}}(a'(x_i))|_2^2$. This suggests a potential attempt at non-contrastive SSL, as it reduces the requirement of two positive augments to only one.

Furthermore, we can derive that the global minimizer of $\mathcal{R}_c$ implies the dynamics in Eq. (2.4), where the representation function $f_\Theta(x)$ is iteratively replaced by the mean representation of its neighbors. This result highlights the implicit relationship established by SimXIR between the sample $x$ and its neighborhood $\mathcal{N}(x)$, which will be thoroughly explored and dissected in the subsequent section.

## 4 THEORETICAL UNDERSTANDING OF NEIGHBOR-AVERAGING DYNAMICS

This section focuses on the theoretical investigation of the implicit bias of neighbor-averaging dynamics on self-supervised representations. We begin by defining the concept of neighbor-connected groups and assuming that the input space comprises disjoint neighbor-connected groups. We then prove that the concise dynamics in Eq. (2.4) exhibit a bias towards the contraction of neighbor-connected groups and the alienation between disjoint groups, which helps prevent collapsed solutions. This is the first work to identify these properties of non-contrastive SSL. Given the common use of $\ell_2$ normalization in the field, we also analogously investigate the behavior of SimXIR in the spherical constrained case where representations are $\ell_2$-normalized. **All proofs can be found in Appendix B.**

### 4.1 NEIGHBOR-CONNECTED GROUP

The fundamental data property we leverage is the neighboring connection of data within the same group (which includes but is not limited to "class" or "cluster"). The main intuition is that, while a random pair of samples from the same group may be far apart, they would be connected by sequences of samples (including raw samples and their augmentations) that are referred to as neighbors within the same group (see Fig. 2). Formally, we define the concept of neighbor-connected group as follows:

**Definition 4.1** (Neighbor-Connected Group). *We call an implicit group $\mathcal{G}$ neighbor-connected, if any two points in $\mathcal{G}$ can be connected with a neighboring path, i.e., $\forall x, y \in \mathcal{G}$, $\exists m \in \mathbb{N}$, $\exists x_1, ..., x_m \in \mathcal{G}$, such that $x_1 \in \mathcal{N}(x)$, $x_2 \in \mathcal{N}(x_1)$, ..., $x_m \in \mathcal{N}(x_{m-1})$, and $y \in \mathcal{N}(x_m)$.*

The structural property of neighbor-connected groups, namely the neighboring path, can be attributed to the rich augmentation techniques in SSL, including Gaussian blur, image solarization, color distortion, and random cropping, etc (Chen et al., 2020; Chen and He, 2021). These augmentations

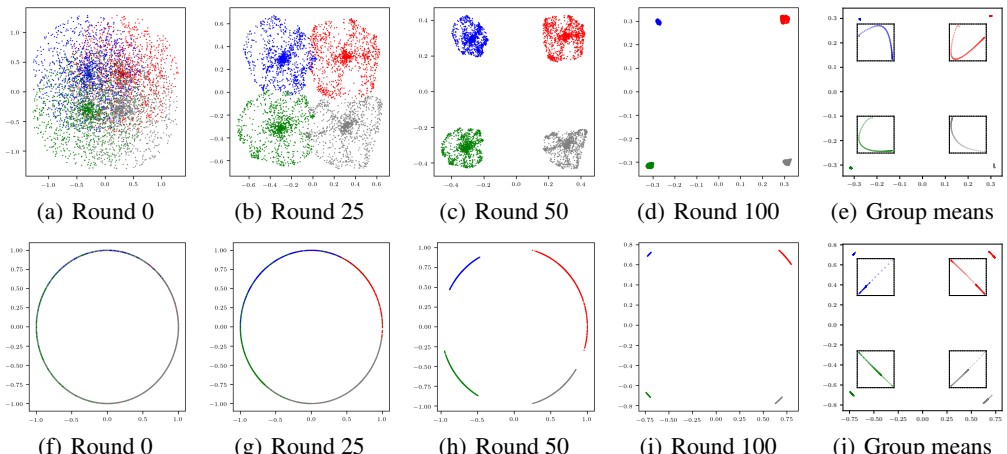

Figure 4: Illustration of the behavior of the representation function (a-d, f-i) and group means (e,j) following Eq. (2.4) (a-e) and Eq. (4.2) (f-j). We divide the input space to four neighbor-connected groups, where each group's samples are randomly generated and their representations are initialized as random points (ensuring separated group means). In plots (e) and (j), the black boxes represent the zoomed-in views. For both unconstrained and spherical constrained cases, the representations within each group gradually contract, tending to collapse into a single point (as shown in Theorem 4.3 and 4.6). Meanwhile, the distance between groups increases (as per Theorem 4.4 and 4.7) while the movement of group means remains essentially unchanged (as hypothesized in Conjecture 4.5).

significantly enrich the diversity of samples from the initially sparse input space, making it possible to connect and organize samples into cohesive groups. For brevity, we can define the input space $\mathcal{X}$ as the set of original samples and their neighbors, with the majority of these neighbors being augmented views. Coincidentally, prior works have proposed the similar concepts, such as expansion property (Wei et al., 2021) and augmentation graph (HaoChen et al., 2021). These concepts depend on the assumption of data continuity and serve to elucidate the interconnectedness of augmented samples.

Accordingly, we provide the separation assumption, which states that the input space is separable:

**Assumption 4.2** (Separation). *We assume that the input space $\mathcal{X}$ is composed of disjoint neighbor-connected groups, i.e., $\mathcal{X} = \cup_{i=1}^{k} \mathcal{G}_i$, where $\mathcal{G}_i$ is neighbor-connected, and $\mathcal{G}_i \cap \mathcal{G}_j = \emptyset, \forall i \neq j$.*

Assumption 4.2 extends the traditional instance-based separability to the new group-based separability, and thus provides a broader perspective for the subsequent theoretical analysis. Due to the absence of human annotations, the formation of neighbor-connected groups is attributed to data characteristics rather than the semantic concept of class, and the number of such groups may far exceed class number. In the following, we will theoretically investigate some intriguing properties of SimXIR.

## 4.2 PROPERTIES OF SIMXIR

In this subsection, we theoretically prove that the neighbor-averaging dynamics of SimXIR in Eq. (2.4) exhibit two intriguing properties: contraction and alienation, which also hold in commonly-used spherical constrained cases. Additionally, we conjecture that the average representation of each group is nearly-invariant during SimXIR's iterations. These properties shed light on the underlying properties of non-contrastive SSL and have the potential to contribute to boosted representations.

### 4.2.1 CONTRACTION OF NEIGHBOR-CONNECTED GROUPS

Non-contrastive SSL that only aligns positive pairs reduces the distance between a sample and its neighbors to capture the underlying invariance to transformations or distortions (Von Kügelgen et al., 2021; Huang et al., 2022; He et al., 2022; Tomasev et al., 2022; Kong and Zhang, 2023). Intuitively, the reduction in distance between the representations in a neighborhood $\mathcal{N}(x)$ will expand to $\mathcal{N}^{(2)}(x), \mathcal{N}^{(3)}(x),...,$ and eventually contracting the whole group $\mathcal{G}$, *i.e.*, resulting to a contraction of the representation space $\{f_{\Theta}(x) : x \in \mathcal{G}\}$. Specifically, we formalize the contraction

Table 1: Top-1 accuracy (%) under k-nearest neighbors classification (where cosine / Euclidean distances are used as metric) on benchmark datasets SVHN and CIFAR-10/-100. Red and blue numbers denote positive gain and negative loss after self-supervised fine-tuning by SimXIR.

| Method | Number of Neighbors $k$ | | | Number of Neighbors $k$ | | | Number of Neighbors $k$ | | |
| | 5 | 11 | 21 | 5 | 11 | 21 | 5 | 11 | 21 |
|---|---|---|---|---|---|---|---|---|---|
| Dataset | SVHN | | | CIFAR-10 | | | Results on CIFAR-100 | | |
| MoCo | 88.675 / 87.827 | 89.467 / 88.833 | 89.813 / 89.451 | 85.36 / 83.28 | 85.65 / 84.08 | 85.63 / 83.97 | 51.33 / 48.48 | 51.94 / 50.60 | 52.89 / 51.18 |
| MoCo* | 91.714 / 91.253 | 92.114 / 92.048 | 92.356 / 92.333 | 85.85 / 84.08 | 86.00 / 84.70 | 85.82 / 85.09 | 51.62 / 49.26 | 53.33 / 51.75 | 53.13 / 51.86 |
| | +3.039 / +3.426 | +2.647 / +3.215 | +2.543 / +2.882 | +0.49 / +0.80 | +0.35 / +0.62 | +0.19 / +1.12 | +0.29 / +0.78 | +1.39 / +1.15 | +0.24 / +0.68 |
| SimCLR | 90.377 / 90.201 | 90.934 / 90.927 | 91.103 / 91.230 | 85.24 / 84.39 | 85.92 / 85.16 | 85.91 / 85.00 | 51.48 / 49.11 | 52.62 / 52.19 | 53.13 / 52.35 |
| SimCLR* | 92.221 / 91.968 | 92.694 / 92.659 | 92.878 / 92.832 | 85.38 / 84.87 | 86.01 / 85.34 | 85.95 / 85.30 | 51.58 / 49.58 | 53.13 / 51.56 | 53.29 / 52.54 |
| | +1.844 / +1.767 | +1.760 / +1.732 | +1.775 / +1.602 | +0.14 / +0.48 | +0.09 / +0.18 | +0.04 / +0.30 | +0.10 / +0.47 | +0.51 / -0.63 | +0.16 / +0.19 |
| BYOL | 91.134 / 90.927 | 91.372 / 91.426 | 91.537 / 91.737 | 86.84 / 84.88 | 87.15 / 85.73 | 86.99 / 86.17 | 51.86 / 50.16 | 53.45 / 52.16 | 53.64 / 52.30 |
| BYOL* | 92.986 / 92.840 | 93.212 / 93.339 | 93.201 / 93.397 | 87.18 / 85.49 | 87.43 / 86.52 | 87.12 / 86.51 | 53.21 / 51.00 | 54.48 / 53.16 | 54.70 / 53.70 |
| | +1.852 / +1.913 | +1.840 / +1.913 | +1.664 / +1.660 | +0.34 / +0.61 | +0.28 / +0.79 | +0.13 / +0.34 | +1.35 / +0.84 | +1.03 / +1.00 | +1.06 / +1.40 |
| SimSiam | 86.056 / 85.871 | 86.832 / 86.782 | 87.197 / 86.962 | 85.34 / 84.08 | 85.48 / 84.92 | 85.47 / 85.15 | 46.71 / 45.18 | 47.81 / 47.14 | 48.09 / 47.39 |
| SimSiam* | 89.344 / 89.252 | 90.035 / 89.828 | 90.208 / 90.147 | 85.35 / 84.35 | 85.69 / 85.22 | 85.65 / 85.42 | 46.73 / 45.33 | 48.45 / 47.91 | 48.80 / 48.41 |
| | +3.288 / +3.381 | +3.203 / +3.046 | +3.011 / +3.185 | +0.01 / +0.27 | +0.21 / +0.30 | +0.18 / +0.27 | +0.02 / +0.15 | +0.64 / +0.77 | +0.71 / +1.02 |

phenomenon as a decrease in the diameter $\sup_{x,y \in \mathcal{G}} \|f_\Theta(x) - f_\Theta(y)\|_2$. We theoretically prove that the representations extracted by $f_\Theta(\cdot)$ on any neighbor-connected group will be contracted during iterations of neighbor-averaging representation replacement.

**Theorem 4.3** (Contraction of Neighbor-Connected Groups). *For any neighbor-connected group $\mathcal{G} \subseteq \mathbb{R}^{d_1}$ and any bounded representation function $f_{\Theta_0} : \mathbb{R}^{d_1} \to \mathbb{R}^{d_2}$, considering the dynamics in Eq. (2.4), we have $\sup_{x,y \in \mathcal{G}} \|f_{\Theta_t}(x) - f_{\Theta_t}(y)\|_2 < \sup_{x,y \in \mathcal{G}} \|f_{\Theta_{t-1}}(x) - f_{\Theta_{t-1}}(y)\|_2$ unless $\sup_{x,y \in \mathcal{G}} \|f_{\Theta_{t-1}}(x) - f_{\Theta_{t-1}}(y)\|_2 = 0$.*

Theorem 4.3 shows that if each sample in a group has its representation replaced by the mean representation of its neighboring samples, then the representations within this group will contract or even converge to a single point. This contraction property provides valuable insights into the understanding of the gathering of representations within groups and also has important implications for improving downstream classification tasks in which compact representations are often desirable (Wang and Isola, 2020; Papyan et al., 2020; HaoChen et al., 2021; Zhou et al., 2022). As empirically proven in Figure 4(a-e), the representations of each group eventually contract into a smaller region and are more discriminative from other groups with an obvious margin.

### 4.2.2 ALIENATION BETWEEN DISJOINT GROUPS

As aforementioned, the contraction of neighbor-connected groups is somewhat analogous to the desired property of intra-class compactness in classification. In this regard, we propose a notion similar to maximizing inter-class separability, *i.e.*, the alienation of any two disjoint neighbor-connected groups, which is formulated as the increase in the minimal distance $\inf_{x_1 \in \mathcal{G}_1, x_2 \in \mathcal{G}_2} \|f_\Theta(x_1) - f_\Theta(x_2)\|_2$ between any two disjoint groups $\mathcal{G}_1$ and $\mathcal{G}_2$. The formal description is as follows:

**Theorem 4.4** (Alienation Between Disjoint Groups). *For any two different neighbor-connected groups $\mathcal{G}_1, \mathcal{G}_2 \subseteq \mathbb{R}^{d_1}$ and any bounded representation function $f_{\Theta_0} : \mathbb{R}^{d_1} \to \mathbb{R}^{d_2}$, considering the dynamics in Eq. (2.4), if $\inf_{x_1 \in \mathcal{G}_1, x_2 \in \mathcal{G}_2} \|f_{\Theta_0}(x_1) - f_{\Theta_0}(x_2)\|_2 > 0$, then we have $\inf_{x_1 \in \mathcal{G}_1, x_2 \in \mathcal{G}_2} \|f_{\Theta_t}(x_1) - f_{\Theta_t}(x_2)\|_2 > \inf_{x_1 \in \mathcal{G}_1, x_2 \in \mathcal{G}_2} \|f_{\Theta_{t-1}}(x_1) - f_{\Theta_{t-1}}(x_2)\|_2$ unless $\sup_{x,y \in \mathcal{G}_1} \|f_{\Theta_{t-1}}(x) - f_{\Theta_{t-1}}(y)\|_2 = \sup_{x,y \in \mathcal{G}_2} \|f_{\Theta_{t-1}}(x) - f_{\Theta_{t-1}}(y)\|_2 = 0$.*

In Theorem 4.4, the condition $\inf_{x_1 \in \mathcal{G}_1, x_2 \in \mathcal{G}_2} \|f_{\Theta_0}(x_1) - f_{\Theta_0}(x_2)\|_2 > 0$ meaning that the initialized representations of $\mathcal{G}_1$ and $\mathcal{G}_2$ are disjoint. It seems to be strict, but makes sense in conjunction with Theorem 4.3, because there may exist a $t' \in \mathbb{N}^+$ that satisfies $\inf_{x_1 \in \mathcal{G}_1, x_2 \in \mathcal{G}_2} \|f_{\Theta_{t'}}(x_1) - f_{\Theta_{t'}}(x_2)\|_2 > 0$ while Theorem 4.3 ensures that the representations within each group are always contracting. The alienation between disjoint groups intuitively plays a vital role in the separability between groups and also partly prevents all representations from collapsing to the same vector.

### 4.2.3 NEAR INVARIANCE OF GROUP MEANS

According to the dynamics in Eq. (2.4), we can informally state that, for any neighbor-connected group $\mathcal{G}_i$, its mean representation $\mathbb{E}_{x \in \mathcal{G}} f_\Theta(x)$ would be nearly-invariant:

**Conjecture 4.5** (Near Invariance of Group Means). *For any bounded representation function $f_{\Theta_0}$ : $\mathbb{R}^{d_1} \to \mathbb{R}^{d_2}$, considering the dynamics $\forall x \in \mathcal{X}, f_{\Theta_t}(x) = \mathbb{E}_{x' \in \mathcal{N}(x)} f_{\Theta_{t-1}}(x')$ for $t \in \mathbb{N}^+$, we have*

$$\mu_i^t = \mathbb{E}_{x \in \mathcal{G}_i} f_{\Theta_t}(x) = \mathbb{E}_{x \in \mathcal{G}_i} \mathbb{E}_{x' \in \mathcal{N}(x)} f_{\Theta_{t-1}}(x') \approx \mathbb{E}_{x \in \mathcal{G}_i} f_{\Theta_{t-1}}(x) = \mu_i^{t-1}. \quad (4.1)$$

The above conjecture suggests that these group means could serve as anchors for their contraction. As illustrated in Fig. 4(e), despite the fact that the mean representations of each group are dynamic, their movement is nearly unchanged. This has significant practical implications for model initialization to avoids collapsed solutions: models with random weight initialization may make group means close to each other, and as a result, SimXIR could not learn useful representations from a randomly initialized model. Accordingly, we do not learn SimXIR from scratch, but use it to fine-tune off-the-shelf SSL models, which is also referred to as *self-supervised fine-tuning*.

### 4.3 PROPERTIES IN SPHERICAL CONSTRAINED CASES

In SSL, a commonly-used operation is to perform $\ell_2$-normalization to constrain representations on the unit sphere, *i.e.*, $f_\Theta(x) = \frac{h_\Theta(x)}{\|h_\Theta(x)\|_2}$ (where $h_\Theta(\cdot)$ denotes the original representation model), and the objective of $\ell_2$ loss becomes maximizing the cosine similarity between online and target representations. The dynamics of SimXIR in spherical constrained cases can be formulated as

$$f_{\Theta_t}(x) = \mathbb{E}_{x' \in \mathcal{N}(x)} f_{\Theta_{t-1}}(x') / \|\mathbb{E}_{x' \in \mathcal{N}(x)} f_{\Theta_{t-1}}(x')\|_2, \quad (4.2)$$

which is elaborated in supplementary materials. As shown in Figure 4(g-l), each group on the circle obviously exhibits the properties of contraction and alienation. We proceed to prove these two properties in spherical constrained cases:

**Theorem 4.6** (Contraction in Spherical Constrained Cases). *For any neighbor-connected Group $\mathcal{G} \subseteq \mathbb{R}^{d_1}$ and any bounded function $f_{\Theta_0} : \mathbb{R}^{d_1} \to \mathbb{S}^{d_2-1}$, considering the dynamics in Eq. (4.2) if the angle $\angle(f_{\Theta_{t-1}}(x), f_{\Theta_{t-1}}(y)) < 180°$ for any $x, y \in \mathcal{G}$, then we have $\sup_{x,x' \in \mathcal{G}} \|f_{\Theta_t}(x) - f_{\Theta_t}(x')\|_2 < \sup_{x,x' \in \mathcal{G}} \|f_{\Theta_{t-1}}(x) - f_{\Theta_{t-1}}(x')\|_2$ unless $\sup_{x,y \in \mathcal{G}} \|f_{\Theta_{t-1}}(x) - f_{\Theta_{t-1}}(y)\|_2 = 0$.*

As can be seen, Theorem 4.6 requires the angle $\angle(f_{\Theta_{t-1}}(x), f_{\Theta_{t-1}}(y)) < 180°$ for any $x, y \in \mathcal{G}$. We can achieve this requirement by incorporating a ReLU layer (Nair and Hinton, 2010) before applying the $\ell_2$ normalization to guarantee the contraction of neighbor-connected groups. In fact, this is the commonly used setting in ResNet (He et al., 2016). The alienation property in spherical constrained cases can also be proved:

**Theorem 4.7** (Alienation in Spherical Constrained Cases). *For any two disjoint neighbor-connected*

Table 2: Top-1/-5 accuracy (%) under linear evaluation on SVHN and CIFAR-10/-100, where $*$ and $\dagger$ denote SimXIR (see Eq. (3.1)) and its variant (see Eq. (3.2)), respectively. Red and blue numbers denote positive gain and negative loss after self-supervised fine-tuning by SimXIR and its variant, respectively.

| Methods | Datasets | | |
|---|---|---|---|
| | SVHN | CIFAR-10 | CIFAR-100 |
| MoCo | 92.094 / 99.024 | 86.82 / 99.54 | 54.77 / 83.90 |
| MoCo* | 92.179 / 99.082 | 86.83 / 99.58 | 55.51 / 84.07 |
| | +0.085 / +0.058 | +0.01 / +0.04 | +0.74 / +0.17 |
| MoCo† | 94.004 / 99.282 | 87.01 / 99.57 | 56.37 / 84.74 |
| | +1.910 / +0.258 | +0.19 / +0.03 | +1.60 / +0.84 |
| SimCLR | 92.340 / 99.101 | 86.96 / 99.54 | 54.49 / 83.19 |
| SimCLR* | 92.494 / 99.163 | 87.13 / 99.52 | 54.68 / 83.60 |
| | +0.154 / +0.062 | +0.17 / -0.02 | +0.19 / +0.41 |
| SimCLR† | 94.007 / 99.362 | 87.04 / 99.61 | 55.47 / 84.05 |
| | +1.667 / +0.261 | +0.08 / +0.07 | +0.98 / +0.86 |
| BYOL | 90.681 / 98.644 | 86.79 / 99.38 | 51.85 / 79.58 |
| BYOL* | 91.679 / 98.794 | 87.20 / 99.43 | 53.00 / 80.37 |
| | +0.998 / +0.150 | +0.41 / +0.05 | +1.15 / +0.79 |
| BYOL† | 93.135 / 99.070 | 87.34 / 99.46 | 53.83 / 81.15 |
| | +2.454 / +0.418 | +0.54 / +0.08 | +2.04 / +1.60 |
| SimSiam | 87.243 / 98.402 | 85.58 / 99.45 | 44.29 / 75.64 |
| SimSiam* | 88.530 / 98.767 | 85.95 / 99.53 | 45.74 / 76.16 |
| | +1.287 / +0.365 | +0.37 / +0.08 | +1.45 / +0.52 |
| SimSiam† | 91.380 / 99.059 | 86.39 / 99.56 | 46.50 / 77.33 |
| | +4.137 / +0.657 | +0.81 / +0.11 | +2.21 / +1.69 |

*groups $\mathcal{G}_1, \mathcal{G}_2 \subset \mathbb{R}^{d_1}$ and any bounded representation function $f_{\Theta_0} : \mathbb{R}^{d_1} \to \mathbb{R}^{d_2}$, considering the dynamics in Eq. (4.2), if $\inf_{x_1 \in \mathcal{G}_1, x_2 \in \mathcal{G}_2} \|f_{\Theta_0}(x_1) - f_{\Theta_0}(x_2)\|_2 > 0$, then we have $\inf_{x_1 \in \mathcal{G}_1, x_2 \in \mathcal{G}_2} \|f_{\Theta_t}(x_1) - f_{\Theta_t}(x_2)\|_2 > \inf_{x_1 \in \mathcal{G}_1, x_2 \in \mathcal{G}_2} \|f_{\Theta_{t-1}}(x_1) - f_{\Theta_{t-1}}(x_2)\|_2$ unless $\sup_{x,y \in \mathcal{G}_1} \|f_{\Theta_{t-1}}(x) - f_{\Theta_{t-1}}(y)\|_2 = \sup_{x,y \in \mathcal{G}_2} \|f_{\Theta_{t-1}}(x) - f_{\Theta_{t-1}}(y)\|_2 = 0.*

## 5 EXPERIMENTS

In this section, we provide extensive experiments on downstream linear classification and k-nearest neighbors classification tasks to verify the effectiveness of SimXIR for fine-tuning self-supervised models pretrained on the small datasets SVHN (Netzer et al., 2011) and CIFAR-10/-100 (Krizhevsky, 2009), and a large-scale dataset ImageNet ILSVRC-2012 dataset (Deng et al., 2009b). **More experimental results and details can be found in Appendix C.**

Table 3: Top-1/-5 accuracy (%) under linear evaluation on ImageNet, where $*$ denotes the fine-tuned models by SimXIR. Red numbers denote positive gain after by SimXIR.

| Methods | Top-1/-5 linear probe |
|---------|------------------------|
| Relative Location **Relative Location**$*$ (Doersch et al., 2015) | 34.618 / 59.140 **40.096 / 64.706** +5.478 / +5.566 |
| Rotation prediction **Rotation prediction**$*$ (Komodakis and Gidaris, 2018) | 37.898 / 63.236 **41.416 / 66.218** +3.518 / +2.982 |
| ODC **ODC**$*$ (Zhan et al., 2020) | 57.048 / 80.630 **57.574 / 80.964** +0.526 / +0.334 |
| NPID **NPID**$*$ (Wu et al., 2018) | 57.908 / 80.870 **58.296 / 81.194** +0.388 / +0.324 |
| SimCLR **SimCLR**$*$ (Chen et al., 2020) | 62.560 / 84.878 **62.676 / 84.958** +0.116 / +0.080 |
| MoCo **MoCo**$*$ (He et al., 2020) | 67.592 / 88.128 **67.630 / 88.176** +0.038 / +0.048 |
| SimSiam **SimSiam**$*$ (Chen and He, 2021) | 69.844 / 89.372 **69.954 / 89.408** +0.110 / +0.036 |
| BarlowTwins **BarlowTwins**$*$ (Zbontar et al., 2021) | 71.060 / 89.838 **71.314 / 89.998** +0.254 / +0.160 |
| BYOL **BYOL**$*$ (Grill et al., 2020) | 71.706 / 90.490 **71.816 / 90.502** +0.110 / +0.012 |

**Implementation details.** For image augmentations, we use the same setting as SimSiam (Chen and He, 2021). For model architecture, we use a convolution residual network (He et al., 2016) with 18 layers (replacing the default $7 \times 7$ convolutional preprocessing layer with $3 \times 3$ convolutions) and 50 layers as our base parametric models for small datasets and ImageNet, respectively. For self-supervised models on small datasets, we reproduce MoCo (He et al., 2020), SimCLR (Chen et al., 2020), BYOL (Grill et al., 2020), and SimSiam (Chen and He, 2021) by training ResNet-18 with a cosine decay learning rate schedule (Loshchilov and Hutter, 2017) over 800 epochs. For ImageNet, we directly use off-the-shelf SSL models available at GitHub [1]. We then fine-tune these self-supervised models via SimXIR in only one round with 20 epochs, *i.e.*, $T = 1$, $K = 20N/B$, where $N$ is the number of samples. Inspired by Dong *et al.* (Dong et al., 2022), we adopt a quite small learning rate for fine-tuning, where the learning rates are $5e - 4$ and $5e - 5$ on small datasets and ImageNet, respectively. The weight decay is set to $1e - 6$ across all datasets. We fine-tune self-supervised models on small datasets using a batch size of 512 with a GPU, and on ImageNet using a batch size of 256 split over 8 GPUs.

**Linear evaluation.** To assess the linear separability of the representations generated by SimXIR, we train a linear classifier on top of the frozen representations, following the procedure described in SimSiam (Chen and He, 2021) where we use the LARS optimizer (You et al., 2017). We report top-1 and top-5 accuracy of off-the-shelf SSL models without and with the proposed self-supervised fine-turning in Tables 2 and 3, where SimXIR (see Eq. (3.1)) and its variant (see Eq. (3.2)) are tested. The results demonstrate that self-supervised fine-tuning by SimXIR can boost visual representations over famous SSL models in most cases across both small and large datasets (except the case of fine-tuning SimCLR on CIFAR-10). Notably, the variant of SimXIR delivers significant improvements on SVHN and CIFAR-100.

**k-nearest neighbors (kNN) evaluation.** To further evaluate the effectiveness of SimXIR, we employ a kNN classifier on top of the frozen representation network, using the cosine distance $d_{\cos}(x, y) = 2 - \frac{2\langle x, y \rangle}{\|x\|_2 \|y\|_2}$ and the Euclidean distance $d_{\mathrm{Euc}}(x, y) = \|x - y\|_2$. Table 1 shows the top-1 accuracy (cosine/Euclidean) obtained with kNN classifiers on SVHN and CIFAR-10/-100. The results demonstrate that self-supervised fine-tuning by SimXIR can consistently improve the visual representations initialized by famous SSL models in terms of kNN accuracy with cosine distance metric, regardless of the number of neighbors. Similar results are achieved in terms of kNN accuracy with Euclidean distance metric in most cases except the case of fine-tuning SimCLR on CIFAR-100.

## 6 CONCLUSION

In this paper, we investigate the implicit bias of non-contrastive learning with a concise framework—SimXIR. Based on SimXIR, we for the first time reveals that explicitly aligning positive pairs involves a underlying mechanism to prevent collapsed solutions, which we describe as *contraction of neighbor-connected groups* and *alienation between disjoint groups*. These two properties resemble intra-class compactness and inter-class separability in classification, and help explain why non-contrastive learning can prevent collapsed solutions. We also show that SimXIR can practically serve as self-supervised fine-tuning module that is a novel step for self-supervised pretraining. Empirical results demonstrate the effectiveness of SimXIR.

---

[1] https://github.com/open-mmlab/mmselfsup/tree/master

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

# Appendix for "Towards Understanding Contraction and Alienation in Non-Contrastive Learning with Mean Representation Replacement"

---

**Algorithm 1** SimXIR: Self-supervised fine-tuning algorithm.

---

**Input:** set of samples $\mathcal{X}$, pretrained representation encoder $f_{\Theta_0}$, optimizer optimizer that updates parameters $\Theta$, learning rate $\eta$, distribution of transformations $\mathcal{A}$, total number of rounds $T$, the number of optimization steps $K$, and batch size $B$.

1: **for** $t = 1$ **to** $T$ **do**
2:      $\Theta \leftarrow \Theta_{t-1}$           $\triangleright$ Initialize $\Theta$ with previous parameters
3:      **for** $k = 1$ **to** $K$ **do**
4:          $\mathcal{S} \leftarrow \{x_i \sim \mathcal{X}\}$           $\triangleright$ randomly sample a batch of $B$ examples
5:          **for** $x_i \in \mathcal{S}$ **do**
6:              $a, a' \sim \mathcal{A}'$           $\triangleright$ randomly sample two transformations
7:              $\ell_i \leftarrow \|f_\Theta(a(x_i)) - f_{\Theta_{t-1}}(a'(x_i))\|_2^2$           $\triangleright$ compute loss for each $x_i$
8:          **end for**
9:          $\delta\Theta \leftarrow \frac{1}{B} \sum_{i=1}^{B} \partial_\Theta \ell_i$           $\triangleright$ compute the gradient
10:          $\Theta \leftarrow \text{optimizer}(\Theta, \delta\Theta, \eta)$      $\triangleright$ use specific optimizer to update the online parameters
11:      **end for**
12:      $\Theta_t \leftarrow \Theta$           $\triangleright$ update parameters for current round
13: **end for**
     **Output:** encoder $f_{\Theta_T}$

---

## A CLARIFICATION

In this section, we aim to clarify certain concepts provided in the main body.

### A.1 NEIGHBOR-AVERAGING DYNAMICS.

For the neighbor-averaging dynamics in Eq. 2.4, it can be viewed as a neighborhood-based message passing, like affinity propagation (Frey and Dueck, 2007) in clustering and label propagation (Zhu et al., 2003) in semi-supervised learning. For instance, when the representation function $f$ is reduced to the label function, these neighbor-averaging dynamics encompass the iterative process of label propagation. In this context, it can be expressed as $f_t(x) = \sum_{x'} p(x'|x) f_{t-1}(x')$, where $p(x'|x)$ that denotes the probability of $x'$ belonging to the neighborhood of $x$ can be regarded as the edge weight between two vertices $x'$ and $x$.

### A.2 ABOUT SIMXIR IN SPHERICAL CONSTRAINED CASES

We provide the dynamics of SimXIR in spherical constrained cases where the representations are performed with a $\ell_2$ normalization, *i.e.*, $f_\Theta(x) = \frac{h_\Theta(x)}{\|h_\Theta(x)\|_2}$ (where $h_\Theta(\cdot)$ denotes the encoder in existing SSL models). Therefore, the objective of $\ell_2$ loss becomes maximizing the cosine similarity between online and target representations. The global minimizer of the conditional risk $\mathbb{E}_{x' \in \mathcal{N}(x)} \|\frac{h_\Theta(x)}{\|h_\Theta(x)\|_2} - f_{\Theta_{t-1}}(x')\|_2^2$ is derived as

$$
\begin{aligned}
f_{\Theta_t}(x) &= \arg\min_{f_\Theta} \mathbb{E}_{x' \in \mathcal{N}(x)} \left\| \frac{h_\Theta(x)}{\|h_\Theta(x)\|_2} - f_{\Theta_{t-1}}(x') \right\|_2^2 \\
&= \arg\min_{f_\Theta} \left\| \frac{h_\Theta(x)}{\|h_\Theta(x)\|_2} - \mathbb{E}_{x' \in \mathcal{N}(x)} f_{\Theta_{t-1}}(x') \right\|_2^2 \\
&= \frac{\mathbb{E}_{x' \in \mathcal{N}(x)} f_{\Theta_{t-1}}(x')}{\|\mathbb{E}_{x' \in \mathcal{N}(x)} f_{\Theta_{t-1}}(x')\|_2}.
\end{aligned}
\tag{A.1}
$$

In the case of spherical constraints, it is common to implement an $\ell_2$ normalization layer on top of the encoder network or perform $\ell_2$ normalization before calculating the loss. This approach has been widely adopted in many works (Chen et al., 2020; He et al., 2020; Grill et al., 2020; Chen and He, 2021).

### A.3 ABOUT NEAR INVARIANCE OF GROUP MEANS

**Conjecture 4.5** [Near Invariance of Group Means] *For any bounded representation function $f_{\Theta_0}$ : $\mathbb{R}^{d_1} \to \mathbb{R}^{d_2}$, considering the dynamics $\forall x \in \mathcal{X}$, $f_{\Theta_t}(x) = \mathbb{E}_{x' \in \mathcal{N}(x)} f_{\Theta_{t-1}}(x')$ for $t \in \mathbb{N}^+$, we have*

$$\mu_i^t = \mathbb{E}_{x \in \mathcal{G}_i} f_{\Theta_t}(x) = \mathbb{E}_{x \in \mathcal{G}_i} \mathbb{E}_{x' \in \mathcal{N}(x)} f_{\Theta_{t-1}}(x') \approx \mathbb{E}_{x \in \mathcal{G}_i} f_{\Theta_{t-1}}(x) = \mu_i^{t-1}. \tag{A.2}$$

The conjecture above provides implications that group means are likely to be anchored in a small region, causing the representations in each group to contract around these anchors. Additionally, if these group means are discriminative with a large margin, the representations will eventually exhibit good separation.

For the spherical constrained cases, we also *conjecture* that

$$\mu_i^t = \frac{\mathbb{E}_{x \in \mathcal{G}_i} f_{\Theta_t}(x)}{\|\mathbb{E}_{x \in \mathcal{G}_i} f_{\Theta_t}(x)\|_2} = \frac{1}{\|\mathbb{E}_{x \in \mathcal{G}_i} f_{\Theta_t}(x)\|_2} \mathbb{E}_{x \in \mathcal{G}_i} \frac{\mathbb{E}_{x' \in \mathcal{N}(x)} f_{\Theta_{t-1}}(x')}{\|\mathbb{E}_{x' \in \mathcal{N}(x)} f_{\Theta_{t-1}}(x')\|_2} \approx \frac{\mathbb{E}_{x \in \mathcal{G}_i} f_{\Theta_{t-1}}(x)}{\|\mathbb{E}_{x \in \mathcal{G}_i} f_{\Theta_{t-1}}(x)\|_2} = \mu_i^{t-1}. \tag{A.3}$$

As demonstrated empirically in Figure 4(j), the mean representations of each group are dynamic, yet their movements remain relatively unchanged.

## B PROOFS OF THEOREMS

### B.1 PROOF OF THEOREM 4.3

**Theorem 4.3** [Contraction of Neighbor-Connected Groups] *For any neighbor-connected group $\mathcal{G} \subseteq \mathbb{R}^{d_1}$ and any bounded representation function $f_{\Theta_0} : \mathbb{R}^{d_1} \to \mathbb{R}^{d_2}$, considering the dynamics $\forall x \in \mathcal{X}$, $f_{\Theta_t}(x) = \mathbb{E}_{x' \in \mathcal{N}(x)} f_{\Theta_{t-1}}(x')$ for $t \in \mathbb{N}^+$, we have $\sup_{x,y \in \mathcal{G}} \|f_{\Theta_t}(x) - f_{\Theta_t}(y)\|_2 < \sup_{x,y \in \mathcal{G}} \|f_{\Theta_{t-1}}(x) - f_{\Theta_{t-1}}(y)\|_2$ unless $\sup_{x,y \in \mathcal{G}} \|f_{\Theta_{t-1}}(x) - f_{\Theta_{t-1}}(y)\|_2 = 0$.*

*Proof.* First, we obviously have the fact that if there exists a $t \in \mathbb{N}^+$ that satisfies $\sup_{x,y \in \mathcal{G}} \|f_{\Theta_t}(x) - f_{\Theta_t}(y)\|_2 = 0$, then $\sup_{x,y \in \mathcal{G}} \|f_{\Theta_{t'}}(x) - f_{\Theta_{t'}}(y)\|_2 = 0$ for all $t' \geq t$.

In addition, according to the Cauchy-Schwarz inequality, we can easily derive that

$$\begin{aligned}
&\|f_{\Theta_t}(x) - f_{\Theta_t}(y)\|_2 \\
=&\|\mathbb{E}_{x' \in \mathcal{N}(x)} f_{\Theta_{t-1}}(x') - \mathbb{E}_{y' \in \mathcal{N}(y)} f_{\Theta_{t-1}}(y')\|_2 \\
\leq&\mathbb{E}_{x' \in \mathcal{N}(x)} \mathbb{E}_{y' \in \mathcal{N}(y)} \|f_{\Theta_{t-1}}(x') - f_{\Theta_{t-1}}(y')\|_2 \\
\leq& \sup_{x,y \in D} \|f_{\Theta_{t-1}}(x) - f_{\Theta_{t-1}}(y)\|_2,
\end{aligned} \tag{B.1}$$

where the inequality holds if and only if $f_{\Theta_{t-1}}(x') = f_{\Theta_{t-1}}(x)$ for all $x' \in \mathcal{N}(x)$, $f_{\Theta_{t-1}}(y') = f_{\Theta_{t-1}}(y)$ for all $y' \in \mathcal{N}(y)$, and the pair $(x, y)$ also achieves the supremum $\sup_{x'',y'' \in \mathcal{G}} \|f_{\Theta_{t-1}}(x'') - f_{\Theta_{t-1}}(y'')\|_2$, *i.e.*, the representations for $\mathcal{N}(x)$ and $\mathcal{N}(y)$ collapse to the two ends of the diameter.

As can be seen, the diameter $\sup_{x,y \in \mathcal{G}} \|f_\Theta(x) - f_\Theta(y)\|_2$ of the representation space $\{f_\Theta(x) : x \in \mathcal{G}\}$ is monotonically decreasing when adopting the update strategy $f_{\Theta_t}(x) = \mathbb{E}_{x' \in \mathcal{N}(x)} f_{\Theta_{t-1}}(x')$. In the following, we will prove that the diameter must be reduced unless it becomes 0.

According to Eq. (B.1), for any two points $x^*$ and $y^*$ that satisfy $\|f_{\Theta_t}(x^*) - f_{\Theta_t}(y^*)\|_2 = \sup_{x,y \in \mathcal{G}} \|f_{\Theta_t}(x) - f_{\Theta_t}(y)\|_2 = \sup_{x,y \in \mathcal{G}} \|f_{\Theta_{t-1}}(x) - f_{\Theta_{t-1}}(y)\|_2 > 0$, we obtain

$$\begin{aligned}
f_{\Theta_{t-1}}(x) &= f_{\Theta_{t-1}}(x^*) \text{ for all } x \in \mathcal{N}(x^*), \\
f_{\Theta_{t-1}}(y) &= f_{\Theta_{t-1}}(y^*) \text{ for all } x \in \mathcal{N}(y^*),
\end{aligned} \tag{B.2}$$

and $(x^*, y^*)$ also is the diameter of $\{f_{\Theta_{t-1}}(x) : x \in \mathcal{G}\}$.

The above condition indicates that $\forall x' \in \mathcal{N}(x^*), \forall y' \in \mathcal{N}(y^*)$, $x'$ and $y'$ also form a diameter, *i.e.*, $\|f_{\Theta_t}(x') - f_{\Theta_t}(y')\|_2 = \sup_{x,y\in\mathcal{G}} \|f_{\Theta_t}(x) - f_{\Theta_t}(y)\|_2 = \sup_{x,y\in\mathcal{G}} \|f_{\Theta_{t-1}}(x) - f_{\Theta_{t-1}}(y)\|_2$. And again, we will have $\forall x'' \in \mathcal{N}(x')$, $f_{\Theta_{t-1}}(x'') = f_{\Theta_{t-1}}(x') = f_{\Theta_{t-1}}(x^*)$. Thus, the points in the 2rd order neighborhood $\mathcal{N}^{(2)}(x^*)$ of $x^*$ have the same representation. By analogy, we know that the points in any order neighborhood of $x^*$ also share the same representation $f_{\Theta_{t-1}}(x^*)$. Then we have $\sup_{x,y\in\mathcal{G}} \|f_{\Theta_{t-1}}(x) - f_{\Theta_{t-1}}(y)\|_2 = 0$ since $\mathcal{G}$ is neighbor-connected.

In summary, we have proven that $\sup_{x,y\in\mathcal{G}} \|f_{\Theta_t}(x) - f_{\Theta_t}(y)\|_2 = \sup_{x,y\in\mathcal{G}} \|f_{\Theta_{t-1}}(x) - f_{\Theta_{t-1}}(y)\|_2$ holds if and only if $\sup_{x,y\in\mathcal{G}} \|f_{\Theta_{t-1}}(x) - f_{\Theta_{t-1}}(y)\|_2 = 0$. $\qquad\square$

## B.2 PROOF OF THEOREM 4.4

**Theorem 4.4** [Alienation Between Disjoint Groups] *For any two disjoint neighbor-connected groups $\mathcal{G}_1, \mathcal{G}_2 \subseteq \mathbb{R}^{d_1}$ and any bounded representation function $f_{\Theta_0} : \mathbb{R}^{d_1} \to \mathbb{R}^{d_2}$, considering the dynamics $\forall x \in \mathcal{X}, f_{\Theta_t}(x) = \mathbb{E}_{x'\in\mathcal{N}(x)} f_{\Theta_{t-1}}(x')$ for $t \in \mathbb{N}^+$, if $\inf_{x_1\in\mathcal{G}_1,x_2\in\mathcal{G}_2} \|f_{\Theta_0}(x_1) - f_{\Theta_0}(x_2)\|_2 > 0$, then we have $\inf_{x_1\in\mathcal{G}_1,x_2\in\mathcal{G}_2} \|f_{\Theta_t}(x_1) - f_{\Theta_t}(x_2)\|_2 > \inf_{x_1\in\mathcal{G}_1,x_2\in\mathcal{G}_2} \|f_{\Theta_{t-1}}(x_1) - f_{\Theta_{t-1}}(x_2)\|_2$ unless $\sup_{x,y\in\mathcal{G}_1} \|f_{\Theta_{t-1}}(x) - f_{\Theta_{t-1}}(y)\|_2 = \sup_{x,y\in\mathcal{G}_2} \|f_{\Theta_{t-1}}(x) - f_{\Theta_{t-1}}(y)\|_2 = 0$.*

*Proof.* The proof of this theorem is similar to that of Theorem 3. We have the fact that if there exists a $t \in \mathbb{N}^+$ that satisfies $\sup_{x,y\in\mathcal{G}_1} \|f_{\Theta_{t-1}}(x) - f_{\Theta_{t-1}}(y)\|_2 = \sup_{x,y\in\mathcal{G}_2} \|f_{\Theta_{t-1}}(x) - f_{\Theta_{t-1}}(y)\|_2 = 0$, then $\inf_{x_1\in\mathcal{G}_1,x_2\in\mathcal{G}_2} \|f_{\Theta_{t'}}(x_1) - f_{\Theta_{t'}}(x_2)\|_2 = \inf_{x_1\in\mathcal{G}_1,x_2\in\mathcal{G}_2} \|f_{\Theta_{t'-1}}(x_1) - f_{\Theta_{t'-1}}(x_2)\|_2$ for all $t' \geq t$.

In addition, we have the following inequality

$$
\begin{aligned}
&\|f_{\Theta_t}(x_1) - f_{\Theta_t}(x_2)\|_2 \\
=&\|\mathbb{E}_{x_1'\in\mathcal{N}(x_1)} f_{\Theta_{t-1}}(x_1') - \mathbb{E}_{x_2'\in\mathcal{N}(x_2)} f_{\Theta_{t-1}}(x_2')\|_2 \\
\geq& \inf_{x_1'\in\mathcal{N}(x_1),x_2'\in\mathcal{N}(x_2)} \|f_{\Theta_{t-1}}(x_1') - f_{\Theta_{t-1}}(x_2')\|_2 \\
\geq& \inf_{x_1'\in\mathcal{G}_1,x_2'\in\mathcal{G}_2} \|f_{\Theta_{t-1}}(x_1') - f_{\Theta_{t-1}}(x_2')\|_2,
\end{aligned}
\tag{B.3}
$$

where the inequality holds if and only if $f_{\Theta_{t-1}}(x_1') = f_{\Theta_{t-1}}(x_1)$ for all $x_1' \in \mathcal{N}(x_1)$, $f_{\Theta_{t-1}}(x_2') = f_{\Theta_{t-1}}(x_2)$ for all $x_2' \in \mathcal{N}(x_2)$, and $(x_1, x_2)$ achieve the infimum $\inf_{x_1'\in\mathcal{G}_1,x_2'\in\mathcal{G}_2} \|f_{\Theta_{t-1}}(x_1') - f_{\Theta_{t-1}}(x_2')\|_2$.

As can be seen, the minimal distance between the representation space of $\mathcal{G}_1$ and $\mathcal{G}_2$ will not be reduced when adopting the update strategy $f_{\Theta_t} = \mathbb{E}_{x'\in\mathcal{N}(x)} f_{\Theta_{t-1}}(x')$. Actually, we can prove that the minimal distance must increase, unless they both have diameter 0.

According to Eq. (B.3), for any two points $x_1^* \in \mathcal{G}_1$ and $x_2^* \in \mathcal{G}_2$ that satisfy $\|f_{\Theta_t}(x_1^*) - f_{\Theta_t}(x_2^*)\|_2 = \inf_{x_1\in\mathcal{G}_1,x_2\in\mathcal{G}_2} \|f_{\Theta_t}(x_1) - f_{\Theta_t}(x_2)\|_2 = \inf_{x_1\in\mathcal{G}_1,x_2\in\mathcal{G}_2} \|f_{\Theta_{t-1}}(x_1) - f_{\Theta_{t-1}}(x_2)\|_2$, we have

$$
\begin{aligned}
f_{\Theta_{t-1}}(x_1) &= f_{\Theta_{t-1}}(x_1^*) \text{ for all } x_1 \in \mathcal{N}(x_1^*), \\
f_{\Theta_{t-1}}(x_2) &= f_{\Theta_{t-1}}(x_2^*) \text{ for all } x_2 \in \mathcal{N}(x_2^*),
\end{aligned}
\tag{B.4}
$$

and $(x_1^*, x_2^*)$ achieve the infimum distance between $\{f_{\Theta_{t-1}}(x_1) : x_1 \in \mathcal{G}_1\}$ and $\{f_{\Theta_{t-1}}(x_2) : x_2 \in \mathcal{G}_2\}$.

The above condition indicates that $\forall x_1 \in \mathcal{N}(x_1^*), \forall x_2 \in \mathcal{N}(x_2^*)$, $x_1$ and $x_2$ also achieve the infimum distance, *i.e.*, $\|f_{\Theta_t}(x_1) - f_{\Theta_t}(x_2)\|_2 = \inf_{x_1'\in\mathcal{G}_1,x_2'\in\mathcal{G}_2} \|f_{\Theta_t}(x_1') - f_{\Theta_t}(x_2')\|_2 = \inf_{x_1'\in\mathcal{G}_1,x_2'\in\mathcal{G}_2} \|f_{\Theta_{t-1}}(x_1') - f_{\Theta_{t-1}}(x_2')\|_2$. And again, we will have $\forall x_1' \in \mathcal{N}(x_1)$, $f_{\Theta_{t-1}}(x_1') = f_{\Theta_{t-1}}(x_1) = f_{\Theta_{t-1}}(x_1^*)$, and $\forall x_2' \in \mathcal{N}(x_2)$, $f_{\Theta_{t-1}}(x_2') = f_{\Theta_{t-1}}(x_2) = f_{\Theta_{t-1}}(x_2^*)$. Thus, the points in the 2rd order neighborhoods of $x_1^*$ and $x_2^*$ have the same representation $f_{\Theta_{t-1}(x_1^*)}$ and $f_{\Theta_{t-1}(x_2^*)}$, respectively. By analogy, we know that the points in any order neighborhood of $x_1^*$ share the same representation, and so does $x_2^*$. Then we have $\sup_{x,y\in\mathcal{G}_1} \|f_{\Theta_{t-1}}(x) - f_{\Theta_{t-1}}(y)\|_2 = \sup_{x,y\in\mathcal{G}_2} \|f_{\Theta_{t-1}}(x) - f_{\Theta_{t-1}}(y)\|_2 = 0$ since $\mathcal{G}_1$ and $\mathcal{G}_2$ are neighbor-connected groups.

In summary, we have proven that $\inf_{x_1\in\mathcal{G}_1,x_2\in\mathcal{G}_2} \|f_{\Theta_t}(x_1) - f_{\Theta_t}(x_2)\|_2 = \inf_{x_1\in\mathcal{G}_1,x_2\in\mathcal{G}_2} \|f_{\Theta_{t-1}}(x_1) - f_{\Theta_{t-1}}(x_2)\|_2$ holds if and only if $\sup_{x,y\in\mathcal{G}_1} \|f_{\Theta_{t-1}}(x) - f_{\Theta_{t-1}}(y)\|_2 = \sup_{x,y\in\mathcal{G}_2} \|f_{\Theta_{t-1}}(x) - f_{\Theta_{t-1}}(y)\|_2 = 0$. $\qquad\square$

### B.3 PROOF OF THEOREM 4.6

**Theorem 4.6** *[Contraction of Neighbor-Connected Groups in Spherical Constrained Cases] For any neighbor-connected Group $\mathcal{G} \subseteq \mathbb{R}^{d_1}$ and any bounded function $f_{\Theta_0} : \mathbb{R}^{d_1} \to \mathbb{S}^{d_2-1}$, considering the dynamics $\forall x \in \mathcal{X}$, $f_{\Theta_t}(x) = \frac{\mathbb{E}_{x' \in \mathcal{N}(x)} f_{\Theta_{t-1}}(x')}{\|\mathbb{E}_{x' \in \mathcal{N}(x)} f_{\Theta_{t-1}}(x')\|_2}$ (where $t \in \mathbb{N}^+$ and $\mathbb{E}_{x' \in \mathcal{N}(x)} f_{\Theta_{t-1}}(x') \neq 0$), if the angle $\angle(f_{\Theta_{t-1}}(x), f_{\Theta_{t-1}}(y)) < \pi$ for any $x, y \in \mathcal{G}$, then we have $\sup_{x,x' \in \mathcal{G}} \|f_{\Theta_t}(x) - f_{\Theta_t}(x')\|_2 < \sup_{x,x' \in \mathcal{G}} \|f_{\Theta_{t-1}}(x) - f_{\Theta_{t-1}}(x')\|_2$ unless $\sup_{x,y \in \mathcal{G}} \|f_{\Theta_{t-1}}(x) - f_{\Theta_{t-1}}(y)\|_2 = 0$.*

*Proof.* Considering two points $x^*$ and $y^*$ such that $\|f_{\Theta_t}(x^*) - f_{\Theta_t}(y^*)\|_2 = \sup_{x,y \in \mathcal{G}} \|f_{\Theta_t}(x) - f_{\Theta_t}(y)\|_2$, we have

$$
\begin{aligned}
&\sup_{x,y \in \mathcal{G}} \|f_{\Theta_t}(x) - f_{\Theta_t}(y)\|_2 \\
=& \|f_{\Theta_t}(x^*) - f_{\Theta_t}(y^*)\|_2^2 \\
=& 2 - 2 \cos \text{sim}(f_{\Theta_t}(x^*), f_{\Theta_t}(y^*)) \\
=& 2 - 2 \cos \text{sim} \left( \mathbb{E}_{x \in \mathcal{N}(x^*)} f_{\Theta_{t-1}}(x), \mathbb{E}_{x \in \mathcal{N}(y^*)} f_{\Theta_{t-1}}(y) \right) \\
\leq& 2 - 2 \inf_{x \in \mathcal{N}(x^*), y \in \mathcal{N}(y^*)} \cos \text{sim}(f_{\Theta_{t-1}}(x), f_{\Theta_{t-1}}(y)) \\
\leq& 2 - 2 \inf_{x,y \in \mathcal{G}} \cos \text{sim}(f_{\Theta_{t-1}}(x), f_{\Theta_{t-1}}(y)) \\
=& \sup_{x,y \in \mathcal{G}} \|f_{\Theta_{t-1}}(x) - f_{\Theta_{t-1}}(y)\|_2,
\end{aligned}
\tag{B.5}
$$

where $\cos \text{sim}(x, y)$ denotes the cosine similarity between $x$ and $y$, and the first inequality is based on the fact that the angle $\angle(f_{\Theta_{t-1}}(x), f_{\Theta_{t-1}}(y)) < \pi$ for any $x, y \in \mathcal{G}$. The equality holds if and only if $f_{\Theta_{t-1}}(x) = f_{\Theta_{t-1}}(x^*)$ for all $x \in \mathcal{N}(x^*)$, $f_{\Theta_{t-1}}(y) = f_{\Theta_{t-1}}(y^*)$ for all $y \in \mathcal{N}(y^*)$, and the pair $(x, y)$ also achieves the supremum $\sup_{x',y' \in \mathcal{G}} \|f_{\Theta_{t-1}}(x') - f_{\Theta_{t-1}}(y')\|_2$.

Similar to the proof of Theorem 3, if $\sup_{x,x' \in \mathcal{G}} \|f_{\Theta_t}(x) - f_{\Theta_t}(x')\|_2 = \sup_{x,x' \in \mathcal{G}} \|f_{\Theta_{t-1}}(x) - f_{\Theta_{t-1}}(x')\|_2$, we must have $\sup_{x,y \in \mathcal{G}} \|f_{\Theta_{t-1}}(x) - f_{\Theta_{t-1}}(y)\|_2 = 0$. $\qquad \square$

### B.4 PROOF OF THEOREM 4.7

**Theorem 4.7** [Alienation between Neighbor-Connected Groups in Spherical Constrained Cases] *For any two different neighbor-connected groups $\mathcal{G}_1, \mathcal{G}_2 \subset \mathbb{R}^{d_1}$ and any bounded representation function $f_{\Theta_0} : \mathbb{R}^{d_1} \to \mathbb{R}^{d_2}$, considering the dynamics $\forall x \in \mathcal{X}$, $f_{\Theta_t}(x) = \frac{\mathbb{E}_{x' \in \mathcal{N}(x)} f_{\Theta_{t-1}}(x')}{\|\mathbb{E}_{x' \in \mathcal{N}(x)} f_{\Theta_{t-1}}(x')\|_2}$ (where $t \in \mathbb{N}^+$ and $\mathbb{E}_{x' \in \mathcal{N}(x)} f_{\Theta_{t-1}}(x') \neq 0$), if $\inf_{x_1 \in \mathcal{G}_1, x_2 \in \mathcal{G}_2} \|f_{\Theta_0}(x_1) - f_{\Theta_0}(x_2)\|_2 > 0$, then we have $\inf_{x_1 \in \mathcal{G}_1, x_2 \in \mathcal{G}_2} \|f_{\Theta_t}(x_1) - f_{\Theta_t}(x_2)\|_2 > \inf_{x_1 \in \mathcal{G}_1, x_2 \in \mathcal{G}_2} \|f_{\Theta_{t-1}}(x_1) - f_{\Theta_{t-1}}(x_2)\|_2$ unless $\sup_{x,y \in \mathcal{G}_1} \|f_{\Theta_{t-1}}(x) - f_{\Theta_{t-1}}(y)\|_2 = \sup_{x,y \in \mathcal{G}_2} \|f_{\Theta_{t-1}}(x) - f_{\Theta_{t-1}}(y)\|_2 = 0$.*

*Proof.* Considering two points $x^*$ and $y^*$ such that $\|f_{\Theta_t}(x^*) - f_{\Theta_t}(y^*)\|_2 = \inf_{x \in \mathcal{G}_1, y \in \mathcal{G}_2} \|f_{\Theta_t}(x) - f_{\Theta_t}(y)\|_2$, we have

$$
\begin{aligned}
&\inf_{x \in \mathcal{G}_1, y \in \mathcal{G}_2} \|f_{\Theta_t}(x) - f_{\Theta_t}(y)\|_2 \\
=& \|f_{\Theta_t}(x^*) - f_{\Theta_t}(y^*)\|_2^2 \\
=& 2 - 2 \cos \text{sim}(f_{\Theta_t}(x^*), f_{\Theta_t}(y^*)) \\
=& 2 - 2 \cos \text{sim} \left( \mathbb{E}_{x \in \mathcal{N}(x^*)} f_{\Theta_{t-1}}(x), \mathbb{E}_{x \in \mathcal{N}(y^*)} f_{\Theta_{t-1}}(y) \right) \\
\geq& 2 - 2 \sup_{x \in \mathcal{N}(x^*), y \in \mathcal{N}(y^*)} \cos \text{sim}(f_{\Theta_{t-1}}(x), f_{\Theta_{t-1}}(y)) \\
\geq& 2 - 2 \sup_{x,y \in \mathcal{G}} \cos \text{sim}(f_{\Theta_{t-1}}(x), f_{\Theta_{t-1}}(y)) \\
=& \inf_{x \in \mathcal{G}_1, y \in \mathcal{G}_2} \|f_{\Theta_{t-1}}(x) - f_{\Theta_{t-1}}(y)\|_2,
\end{aligned}
\tag{B.6}
$$

where $\cos \text{sim}(x, y)$ denotes the cosine similarity between $x$ and $y$. The equality holds if and only if $f_{\Theta_{t-1}}(x) = f_{\Theta_{t-1}}(x^*)$ for all $x \in \mathcal{N}(x^*)$, $f_{\Theta_{t-1}}(y) = f_{\Theta_{t-1}}(y^*)$ for all $y \in \mathcal{N}(y^*)$, and the pair $(x, y)$ also achieves the supremum $\sup_{x',y' \in \mathcal{G}} \|f_{\Theta_{t-1}}(x') - f_{\Theta_{t-1}}(y')\|_2$.

Similar to the proof of Theorem 3, if $\inf_{x\in\mathcal{G}_1,y\in\mathcal{G}_2}\|f_{\Theta_t}(x)-f_{\Theta_t}(y)\|_2 = \inf_{x\in\mathcal{G}_1,y\in\mathcal{G}_2}\|f_{\Theta_{t-1}}(x)-f_{\Theta_{t-1}}(y)\|_2$, we must have $\sup_{x,y\in\mathcal{G}_1}\|f_{\Theta_{t-1}}(x)-f_{\Theta_{t-1}}(y)\|_2 = \sup_{x,y\in\mathcal{G}_2}\|f_{\Theta_{t-1}}(x)-f_{\Theta_{t-1}}(y)\|_2 = 0$. $\qquad\square$

## C   EXPERIMENTS

### C.1   TOY EXPERIMENTS

To validate Theorems 4.3, 4.4, 4.6 and 4.7, we conduct toy experiments with the following pseudocode.

Table 4: Pseudocode of SimXIR and its variant for toy experiments in a PyTorch-like style

```
# SimXIR
def toy_simxir(inps, reps, eps=0.2, T=500):
    for t in range(T):
        reps = avg_mean(inps, reps, eps=eps)
    return reps

# The variant of SimXIR
def toy_variant_of_simxir(inps, reps, eps=0.2, T=500):
    for t in range(T):
        reps = cos_mean(inps, reps, eps=eps)
    return reps

# dynamics following $f_{\Theta_{t}}(x)=\E_{x'\in\gN(x)}f_{\Theta_{t-1}}(
                                        x')$
def avg_mean(inps, reps, eps=0.2):
    new_reps = zeros_like(reps)
    for i in range(reps.shape[0]):
        dists = ((inps - inps[i])**2).sum(dim=1).sqrt()
        new_reps[i] = reps[dists<eps].mean(dim=0)
    return new_reps

# dynamics following Equation 1
def cos_mean(inps, reps, eps=0.2):
    new_reps = zeros_like(reps)
    for i in range(reps.shape[0]):
        dists = sqrt(((inps - inps[i])**2).sum(dim=1))
        new_reps[i] = reps[dists<eps].mean(dim=0)
        new_reps[i] = new_reps[i] / (new_reps[i]**2).sum().sqrt()
    return new_reps
```

**Contraction and Alienation Beyond Assumption 2.**    To show what happens when the input space is not separable in Assumption 2, we also conduct toy experiments of illustration of several groups with intersection points in Figure 5. In both unconstrained and spherical constrained cases, the diameter of each group may not decrease continuously due to the presence of intersection inputs. However, it is evident that it is still contracting in a manner that reduces the average pairwise distance within the group (*i.e.*, $\mathbb{E}_{x,y\in\mathcal{G}}\|f_\Theta(x)-f_\Theta(y)\|_2^2$). Similarly, the alienation between groups may obey in the way that increases the average pairwise distance between groups (*i.e.*, $\mathbb{E}_{x_1\in\mathcal{G}_1,x_2\in\mathcal{G}_2}\|f_\Theta(x_1)-f_\Theta(x_2)\|_2^2$). Meanwhile, the movement of group means remains essentially unchanged (as hypothesized in Conjecture 4.5).

### C.2   ON SMALL DATASETS

For the experiments of improving self-supervised models on SVHN (Netzer et al., 2011), CIFAR-10 and CIFAR-100 (Krizhevsky, 2009), we reproduce MoCo (He et al., 2020), SimCLR (Chen et al., 2020), BYOL (Grill et al., 2020), and SimSiam (Chen and He, 2021) by training the backbone ResNet-18 (replacing the default $7\times7$ convolutional preprocessing layer with $3\times3$ convolutions)

with a cosine decay learning rate schedule over 800 epochs. We then utilized SimXIR and its variant to fine-tune these self-supervised models. The details of MoCo, SimCLR, BYOL, and SimSiam are as follows:

- **MoCo.** We utilize the momentum update strategy for our target network, incorporating a moving average decay of 0.999. Additionally, we employ a two-layer MLP as the projector of MoCo, which is composed of [Linear(512, 128), ReLU, Linear(128, 128)]. To calculate the contrastive loss, we set the temperature parameter to $\tau = 0.2$ and incorporate a memory bank with a queue size of 2048.

- **SimCLR.** We use a two-layer MLP as the projector of SimCLR, which is composed of [Linear(512, 512), BatchNorm, ReLU, Linear(512, 128), BatchNorm]. To calculate the contrastive loss, we set the temperature as $\tau = 0.2$.

- **BYOL.** We use a two-layer MLP as the projector of BYOL that is composed of [Linear(512, 128), ReLU, Linear(128, 128)], and another two-layer MLP as the predictor that is composed of [Linear(128, 512), BatchNorm, ReLU, Linear(512, 128)].

- **SimSiam.** We use a three-layer MLP as the projector of BYOL that is composed of [Linear(512, 512), BatchNorm, ReLU, Linear(512, 512), BatchNorm, ReLU, Linear(512, 128), BatchNorm], and another two-layer MLP as the predictor that is composed of [Linear(128, 512), BatchNorm, ReLU, Linear(512, 128)].

## C.3 ON IMAGENET

For experiments on ImageNet (Deng et al., 2009a), we use off-the-shelf self-supervised models available at GitHub[2]. Specifically, we use the pretrained models Relative Location[3] (Doersch et al., 2015), Rotation Location[4] (Komodakis and Gidaris, 2018), ODC[5] (Zhan et al., 2020), NPID[6] (Wu et al., 2018), SimCLR[7] (Chen et al., 2020), MoCo[8] (He et al., 2020), SimSiam[9] (Chen and He, 2021), BarlowTwins[10] (Zbontar et al., 2021), and BYOL[11] (Grill et al., 2020). We then fine-tune these self-supervised models via SimXIR over 20 epochs, using SGD optimizer with a learning rate of 5e-4 and weight decay of 1e-6.

---

[2]https://github.com/open-mmlab/mmselfsup/blob/master/docs/en/model_zoo.md
[3]https://download.openmmlab.com/mmselfsup/relative_loc
[4]https://download.openmmlab.com/mmselfsup/rotation_pred
[5]https://download.openmmlab.com/mmselfsup/odc
[6]https://download.openmmlab.com/mmselfsup/npid
[7]https://download.openmmlab.com/mmselfsup/simclr
[8]https://download.openmmlab.com/mmselfsup/moco
[9]https://download.openmmlab.com/mmselfsup/simsiam
[10]https://download.openmmlab.com/mmselfsup/barlowtwins
[11]https://download.openmmlab.com/mmselfsup/byol

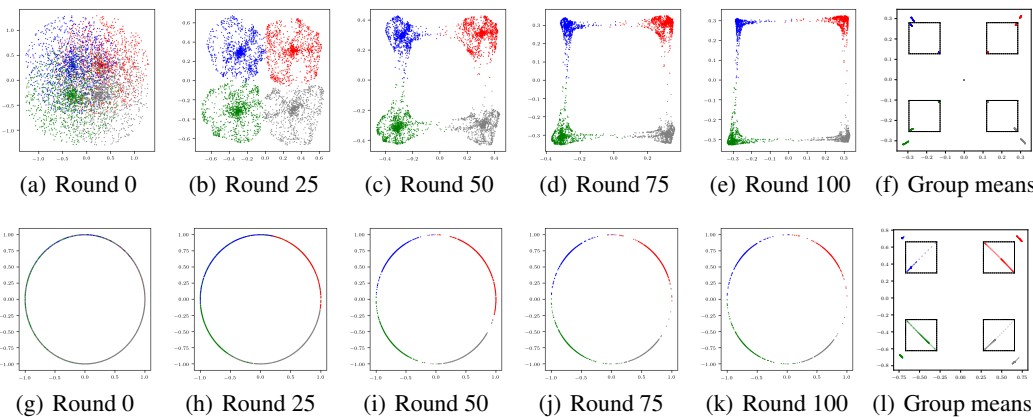

(a) Round 0    (b) Round 25    (c) Round 50    (d) Round 75    (e) Round 100    (f) Group means

(g) Round 0    (h) Round 25    (i) Round 50    (j) Round 75    (k) Round 100    (l) Group means

Figure 5: Illustration of the behavior of the representation function (a-e, g-k) and group means (f, l) following $f_{\Theta_t}(x) = \mathbb{E}_{x' \in \mathcal{N}(x)} f_{\Theta_{t-1}}(x')$ (a-f) and $f_{\Theta_t}(x) = \frac{\mathbb{E}_{x' \in \mathcal{N}(x)} f_{\Theta_{t-1}}(x')}{\|\mathbb{E}_{x' \in \mathcal{N}(x)} f_{\Theta_{t-1}}(x')\|_2}$ (g-l). We divide the input space to four groups in which intersection exists, where each group's samples are randomly generated and their representations are initialized as random points (ensuring separated group means). In plots (f) and (l), the black boxes represent the zoomed-in views. In both unconstrained and spherical constrained cases, the diameter of each group may not decrease continuously due to the presence of intersection inputs. However, it is evident that is still contracting in a manner that reduces the average pairwise distance within the group (*i.e.*, $\mathbb{E}_{x,y \in \mathcal{G}} \|f_\Theta(x) - f_\Theta(y)\|_2^2$). Meanwhile, the movement of group means remains essentially unchanged (as hypothesized in Conjecture 4.5).

Table 5: Top-1 accuracies (in %) under k-nearest neighbors classification (where cosine/Euclidean distances are used as metric) on benchmark datasets SVHN and CIFAR-10/-100 with backbone ResNet-18 He et al. (2016). Results with better performance are **highlighted** compared to baselines. Red and blue numbers denote positive gain and negative loss after self-supervised fine-tuning by SimXIR, respectively.

| Dataset | Method | Number of Neighbors $k$ | | | | | |
|---|---|---|---|---|---|---|---|
| | | 5 | 7 | 11 | 15 | 17 | 21 |
| SVHN | MoCo | 88.675 / 87.827 | 89.067 / 88.287 | 89.467 / 88.833 | 89.670 / 89.232 | 89.670 / 89.209 | 89.813 / 89.451 |
| | **MoCo***  | **91.714 / 91.253** | **91.879 / 91.514** | **92.114 / 92.048** | **92.267 / 92.152** | **92.309 / 92.263** | **92.356 / 92.333** |
| | | +3.039 / +3.426 | +2.812 / +3.227 | +2.647 / +3.215 | +2.597 / +2.920 | +2.639 / +3.054 | +2.543 / +2.882 |
| | SimCLR | 90.377 / 90.201 | 90.638 / 90.523 | 90.934 / 90.927 | 91.007 / 91.126 | 91.073 / 91.192 | 91.103 / 91.230 |
| | **SimCLR*** | **92.221 / 91.968** | **92.409 / 92.375** | **92.694 / 92.659** | **92.859 / 92.682** | **92.832 / 92.778** | **92.878 / 92.832** |
| | | +1.844 / +1.767 | +1.771 / +1.852 | +1.760 / +1.732 | +1.852 / +1.556 | +1.759 / +1.586 | +1.775 / +1.602 |
| | BYOL | 91.134 / 90.927 | 91.380 / 91.165 | 91.372 / 91.426 | 91.507 / 91.580 | 91.457 / 91.591 | 91.537 / 91.737 |
| | **BYOL*** | **92.986 / 92.840** | **93.085 / 93.174** | **93.212 / 93.339** | **93.216 / 93.374** | **93.208 / 93.397** | **93.201 / 93.397** |
| | | +1.852 / +1.913 | +1.705 / +2.009 | +1.840 / +1.913 | +1.709 / +1.794 | +1.751 / +1.806 | +1.664 / +1.660 |
| | SimSiam | 86.056 / 85.871 | 86.382 / 86.332 | 86.832 / 86.782 | 86.970 / 86.901 | 87.185 / 86.897 | 87.197 / 86.962 |
| | **SimSiam*** | **89.344 / 89.252** | **89.763 / 89.609** | **90.035 / 89.828** | **90.066 / 89.993** | **90.158 / 89.947** | **90.208 / 90.147** |
| | | +3.288 / +3.381 | +3.381 / +3.277 | +3.203 / +3.046 | +3.096 / +3.092 | +2.973 / +3.050 | +3.011 / +3.185 |
| CIFAR10 | MoCo | 85.36 / 83.28 | 85.73 / 83.65 | 85.65 / 84.08 | 85.63 / 84.13 | 85.63 / 84.09 | 85.63 / 83.97 |
| | **MoCo*** | **85.85 / 84.08** | **85.74 / 84.57** | **86.00 / 84.70** | **85.84 / 84.67** | **85.94 / 84.92** | **85.82 / 85.09** |
| | | +0.49 / +0.80 | +0.01 / +0.92 | +0.35 / +0.62 | +0.21 / +0.54 | +0.31 / +0.83 | +0.19 / +1.12 |
| | SimCLR | 85.24 / 84.39 | 85.60 / 84.85 | 85.92 / 85.16 | 85.74 / 85.29 | 85.71 / 85.12 | 85.91 / 85.00 |
| | **SimCLR*** | **85.38 / 84.87** | **85.78 / 85.19** | **86.01 / 85.34** | **86.05 / 85.32** | **86.00 / 85.37** | **85.95 / 85.30** |
| | | +0.14 / +0.48 | +0.18 / +0.34 | +0.09 / +0.18 | +0.31 / +0.03 | +0.29 / +0.25 | +0.04 / +0.30 |
| | BYOL | 86.84 / 84.88 | 87.05 / 85.19 | 87.15 / 85.73 | 87.06 / 85.93 | 86.94 / 86.03 | 86.99 / 86.17 |
| | **BYOL*** | **87.18 / 85.49** | **87.25 / 86.15** | **87.43 / 86.52** | **87.20 / 86.54** | **87.06 / 86.48** | **87.12 / 86.51** |
| | | +0.34 / +0.61 | +0.20 / +0.96 | +0.28 / +0.79 | +0.14 / +0.61 | +0.12 / +0.45 | +0.13 / +0.34 |
| | SimSiam | 85.34 / 84.08 | 85.35 / 84.41 | 85.48 / 84.92 | 85.44 / 85.14 | 85.35 / 85.18 | 85.47 / 85.15 |
| | **SimSiam*** | **85.35 / 84.35** | **85.47 / 84.70** | **85.69 / 85.22** | **85.67 / 85.45** | **85.89 / 85.46** | **85.65 / 85.42** |
| | | +0.01 / +0.27 | +0.12 / +0.29 | +0.21 / +0.30 | +0.23 / +0.31 | +0.54 / +0.28 | +0.18 / +0.27 |
| CIFAR100 | MoCo | 51.33 / 48.48 | 51.70 / 49.50 | 51.94 / 50.60 | 52.62 / 51.00 | 52.54 / 51.12 | 52.89 / 51.18 |
| | **MoCo*** | **51.62 / 49.26** | **52.71 / 50.60** | **53.33 / 51.75** | **53.21 / 52.05** | **53.04 / 52.33** | **53.13 / 51.86** |
| | | +0.29 / +0.78 | +1.01 / +1.10 | +1.39 / +1.15 | +0.59 / +1.05 | +0.50 / +1.21 | +0.24 / +0.68 |
| | SimCLR | 51.48 / 49.11 | 52.37 / 50.45 | 52.62 / 52.19 | 52.89 / 52.33 | 52.82 / 52.47 | 53.13 / 52.35 |
| | **SimCLR*** | **51.58 / 49.58** | **52.50 / 50.62** | **53.13** / 51.56 | **53.11** / 52.31 | **53.35 / 52.50** | **53.29 / 52.54** |
| | | +0.10 / +0.47 | +0.13 / +0.17 | +0.51 / -0.63 | +0.22 / -0.02 | +0.53 / +0.03 | +0.16 / +0.19 |
| | BYOL | 51.86 / 50.16 | 52.97 / 51.27 | 53.45 / 52.16 | 53.62 / 52.33 | 53.83 / 52.20 | 53.64 / 52.30 |
| | **BYOL*** | **53.21 / 51.00** | **53.89 / 52.57** | **54.48 / 53.16** | **54.46 / 53.73** | **54.69 / 53.96** | **54.70 / 53.70** |
| | | +1.35 / +0.84 | +0.92 / +1.30 | +1.03 / +1.00 | +0.84 / +1.40 | +0.86 / +1.76 | +1.06 / +1.40 |
| | SimSiam | 46.71 / 45.18 | 47.40 / 46.34 | 47.81 / 47.14 | 48.41 / 47.50 | 48.41 / 47.57 | 48.09 / 47.39 |
| | **SimSiam*** | **46.73 / 45.33** | **47.62 / 46.85** | **48.45 / 47.91** | **48.71 / 48.40** | **48.68 / 48.08** | **48.80 / 48.41** |
| | | +0.02 / +0.15 | +0.22 / +0.51 | +0.64 / +0.77 | +0.30 / +0.90 | +0.27 / +0.51 | +0.71 / +1.02 |

