# OpenReview forum: "Contraction and Alienation: Towards Theoretical Understanding of Non-Contrastive Learning with Neighbor-Averaging Dynamics"
_ICLR.cc/2024/Conference — ICLR 2024 Conference Withdrawn Submission_

### Official Review · Reviewer_A6SE · 2023-10-30

**Soundness:** 4 excellent
**Presentation:** 3 good
**Contribution:** 3 good
**Rating:** 6
**Confidence:** 4

**Summary:**

This paper theoretically understands the reason why the non-contrastive SSL (only with positive pairs alignment) does not collapse. They propose a framwork called SimXIR through the neighbor-averaging dynamics and discover a novel implicite bias of non-contrastive SSL. They also propose the self-supervised fine-tuning, a new fine-tuning paradiam, to enhance the off-the-shelf models.

**Strengths:**

- They propose a new non-contrastive SSL architecture with neighbor-averaging dynamics. This method is more simple and has solid theoretical guarantee of no collapse.

**Weaknesses:**

- The writting needs to be further polished. For example, the term "neighbor-averaging dynamics" appears very early, but they do not give even a intuitive explaination, making the non-expert readers confusing.
- Experiments seem to lack the results of pre-training from scratch by SimXIR. To my understanding, SimXIR is not only a fine-tuning module, but also can serve as a self-supervised pre-training framework. As they have stated, the random initialization may make the group means close to each other, but this case seems very uncommon in practice.
- The comparison between two variants of SimXIR is missing.
- How to justify that the neighbors of a data point are mainly its transformed versions? This assumption misses the discussion about its reasonability. For fine-grained data, if this assumption might be violated? If possible, does the SimXIR still work?
- Some typos. For example, Eq. (3.2)

**Questions:**

- According to Fig. 3, the neighbor-averaging dynamics can boost the existing clustering methods. This result is very interesting. But the details about how to incorporate the neighbor-averaging dynamics into clustering are unclear for me. Please elaborate this problem.
- The neighbor-averaging dynamics can replace the asymmetric structure and achieve good performance. Assume that the asymmetric structure is still adopted together with the neighbor-averaging dynamics. Does this operation boost the performance of existing method such as BYOL?
- Why does SimXIR remove the projection layer? This trick is widely used by current SSL methods. Is the projection layer removed for ease of theretical analysis?
- How does the batch size affect the performance of SimXIR?

---

> ### Author Response · Authors · 2023-11-19
> **Response to Reviewer A6SE**
>
> We thank the reviewer for the constructive comments and insightful suggestions.
>
> **Comment 1:** According to Fig. 3, the neighbor-averaging dynamics can boost the existing clustering methods. This result is very interesting. But the details about how to incorporate the neighbor-averaging dynamics into clustering are unclear for me. Please elaborate this problem.
>
> **Response**: Thank you for your insightful observation regarding the potential of neighbor-averaging dynamics to enhance existing clustering methods, as depicted in Fig. 3. We appreciate your interest in this aspect, and we are happy to provide a more detailed explanation of how the neighbor-averaging dynamics are incorporated into clustering.
>
> The process involves the following steps:
>
> - **Initialization**: Begin with producing initial clustering results using established clustering methods. These initial results serve as the starting point for the label function, denoted as $f_0(x)$.
>
> - **Neighborhood Definition**: Define the neighborhood of each sample $x$ in the feature space, denoted as $\mathcal{N}(x)$. In our approach, we use a simple geometric criterion, considering samples within a distance of 0.4 from $x$.
>
> - **Iterative Label Update**: For each sample $x$, iteratively update its label using the neighbor-averaging dynamics. The label at each iteration $t+1$ is computed as the average of labels from the defined neighborhood: $f_{t+1}(x)=\frac{1}{|\mathcal{N}(x)|}\sum_{x'\in \mathcal{N}(x)} f(x')$.
>
> This process is repeated for a specified number of iterations, allowing the labels to evolve based on the information from neighboring samples.
>
> ---
>
> **Comment 2:** The neighbor-averaging dynamics can replace the asymmetric structure and achieve good performance. Assume that the asymmetric structure is still adopted together with the neighbor-averaging dynamics. Does this operation boost the performance of existing method such as BYOL?
>
> **Response:** Thank you for your insightful comment. Your question about combining the asymmetric structure with neighbor-averaging dynamics to boost existing methods like BYOL is intriguing. When the asymmetric structure is retained alongside the neighbor-averaging dynamics, the operation can indeed be seen as a combination of elements from both paradigms. Specifically, the neighbor-averaging dynamics will be equivalent to the SimSiam framework but the target network is the previous online network without a predictor. When the target network is alternated after a certain number of SGD steps, this framework can be directly used for self-supervised pretraining, and achieve the following linear evaluation accuracy with 100-epoch pre-training, as demonstrated in [1]:
>
> |              | 1-step | 10-step | 100-step | 1-epoch |
> | ------------ | ------ | ------- | -------- | ------- |
> | linear probe | 68.1   | 68.7    | 68.9     | 67.0    |
>
> Here, 1-step is equivalent to SimSiam, and 1-epoch denotes the target network is from the online network in the last epoch.
>
> ---
>
> **Comment 3:** Why does SimXIR remove the projection layer? This trick is widely used by current SSL methods. Is the projection layer removed for ease of theretical analysis?
>
> **Response:** Thank you for your inquiry. The decision to remove the projection layer in SimXIR is for the purpose of facilitating more straightforward theoretical analysis on self-supervised representations. The underlying theory necessitates maintaining symmetry in the structure between the online encoder and the target encoder. While it is worth noting that the projection layer does not necessarily have to be removed, the deliberate exclusion is specifically applied to the asymmetric prediction layer for theoretical considerations.
>
> ---
>
> **Comment 4:** How does the batch size affect the performance of SimXIR?
>
> **Response:** Thank you for your inquiry. In our experiments, SimXIR maintains the same batch size as used during self-supervised pre-training, ensuring consistency in statistics of Batch Normalization throughout the fine-tuning process.
>
> [1] Chen, Xinlei, and Kaiming He. "Exploring simple siamese representation learning." Proceedings of the IEEE/CVF conference on computer vision and pattern recognition. 2021.

---

> ### Author Response · Authors · 2023-11-23
> **Looking forward to your feedback**
>
> Dear reviewer A6SE,
>
> Thanks again for your valuable time and insightful comments. As the deadline for the Author/Reviewer discussion is approaching, it would be nice of you to let us know whether our answers have solved your concerns so that we can better improve our work. We are looking forward to your feedback.
>
> Best regards,
>
> Authors of Paper 222

---

> > ### Comment · Reviewer_A6SE · 2023-11-23
> >
> > Thank you for your detailed clarifications. I have gone through your feedbacks and other reviewers' comments, and my concerns have been well addressed. Overall, I am willing to support this paper.

---

> > > ### Author Response · Authors · 2023-11-23
> > > **Thank you!**
> > >
> > > Thank you for your feedback! We are pleased to address your concerns and greatly appreciate your reviews, which play a crucial role in improving our work.
> > >
> > > Best regards,
> > > The authors

---

### Official Review · Reviewer_UTfC · 2023-10-30

**Soundness:** 3 good
**Presentation:** 2 fair
**Contribution:** 2 fair
**Rating:** 6
**Confidence:** 2

**Summary:**

This paper proposes the SimXIR framework for non-contrastive self-supervised learning. The SimXIR fixes the online network of the last round as the target network to supervise the learning of the current model, which prevents the collapsed solutions without the asymmetric tricks. Theoretical and visualization results show that the SIMXIR compresses the intra-group distances and discriminates the features of the disjoint groups.

**Strengths:**

+ Provide a simple framework for non-contrastive SSL
+ Visualize data distributions to validate theoretical results.
+ Self-supervised fine-tuning significantly improves the performance of popular self-supervised models.

**Weaknesses:**

This paper provides a sufficient mathematical analysis on the proposed SimXIR architecture; however, I have the following concerns:
- Comparison with SimSiam, based on Fig. 1(a), the SimXIR without the MLP predictor and applying the l2 loss to train the model. How do such modifications prevent the model from collapsing during the training process?
- The experimental results are mainly focused on the self-supervised fine-tuning, and the performance of SimXIR working on the randomly initialized models is not presented. In addition, the performance gain of SimXIR on large datasets (such as ImageNet) is insufficient when fine-tuning with the SOTA SSL techniques.
- The numerical results of SimXIR in the tables are evaluated with only one iteration round. The multi-round ablation study is not reported.
- In addition to the accuracy results of self-supervised fine-tuning on the popular datasets and the visualization of the toy experiments, please provide more experimental evidence to validate the theoretical results.
- Typo in Eq. (3.2).

**Questions:**

See weaknesses.

---

> ### Author Response · Authors · 2023-11-19
> **Response to Reviewer UTfC**
>
> We thank the reviewer for the constructive comments and insightful suggestions.
>
> **Comment 1:** *Comparison with SimSiam, based on Fig. 1(a), the SimXIR without the MLP predictor and applying the l2 loss to train the model. How do such modifications prevent the model from collapsing during the training process?* and *The experimental results are mainly focused on the self-supervised fine-tuning, and the performance of SimXIR working on the randomly initialized models is not presented.*
>
> **Response:** Thank you for the kind comment. Throughout the derivation of neighbor-averaging dynamics, we uncovered essential properties, including contraction and alienation, which play a crucial role in preventing collapsed representations without the need for an asymmetric structure.
>
> In our validation process, SimXIR appears to struggle to learn good representations from a randomly initialized model. This limitation can be attributed to the absence of a sufficiently strong initial representation in the randomly initialized model, constraining the potential gains achievable by SimXIR. To rigorously assess the properties of neighbor-averaging dynamics in practical experiments, we chose to employ pre-trained models as the initial models for SimXIR. The experimental results robustly confirm that SimXIR indeed improves representations, with a notable impact on models with weakly initialized performance. In these cases, SimXIR imparts substantial enhancements, further validating the effectiveness of neighbor-averaging dynamics.ations, particularly for models with weakly initialized performance, where SimXIR imparts substantial enhancements.
>
> ---
>
> **Comment 2:** In addition, the performance gain of SimXIR on large datasets (such as ImageNet) is insufficient when fine-tuning with the SOTA SSL techniques.
>
> **Response:** Thank you for your insightful comment. The primary emphasis of our paper is to demonstrate that SimXIR without asymmetric structure, characterized by neighbor-averaging dynamics, can enhance representations based on a favorable initialization. Our experiments are tailored to validate this key assertion. While it is acknowledged that SimXIR shows marginal improvement for models that are well-pretrained and already exhibit strong performance, we emphasize more substantial enhancements for models with initially weaker performance, such as Relative Location and Rotation Prediction. This underscores the efficacy of SimXIR in improving representations even in the absence of asymmetric structure.
>
> ---
>
> **Comment 3:** *The numerical results of SimXIR in the tables are evaluated with only one iteration round. The multi-round ablation study is not reported* and *In addition to the accuracy results of self-supervised fine-tuning on the popular datasets and the visualization of the toy experiments, please provide more experimental evidence to validate the theoretical results.*
>
> **Response:** Thank you for your valuable suggestions. We acknowledge the importance of conducting a multi-round ablation study and providing additional experimental evidence to validate our theoretical results. This will involve a more comprehensive analysis of our work. We will provide these additional experiments that are thoroughly detailed in the revised version.
>
> ---
>
> **Comment 4:** Typo in Eq. (3.2).
>
> **Response:** Thank you for your careful review. We have rectified the mentioned typos and addressed some other potential errors.

---

> ### Author Response · Authors · 2023-11-23
> **Looking forward to your feedback**
>
> Dear reviewer UTfC,
>
> Thanks again for your valuable time and insightful comments. As the deadline for the Author/Reviewer discussion is approaching, it would be nice of you to let us know whether our answers have solved your concerns so that we can better improve our work. We are looking forward to your feedback.
>
> Best regards,
>
> Authors of Paper 222

---

### Official Review · Reviewer_ByEQ · 2023-11-01

**Soundness:** 2 fair
**Presentation:** 3 good
**Contribution:** 2 fair
**Rating:** 5
**Confidence:** 4

**Summary:**

This paper investigates non-contrastive learning from neighbor-averaging dynamics. It propose SimXIR which has good theoretical properties: contraction and alienation. Experimental results show that SimXIR can enhance representations of off-the-shelf SSL models

**Strengths:**

1. The authors give a novel perspective to understand non-contrastive learning methods and show symmetric design can improve the self-supervised learning.
2.  The paper identifies two theoretical properties of non-contrastive learning. These theoretical justifications are novel.
3.  Experimental results verify its effectiveness in boosting representations across various SSL models.

**Weaknesses:**

1. Limited novelty: The method SimXIR looks very similar to Knowledge Distillation（mean teacher): SSL pretrain then knowledge distillation.

2. The gain performance on CIFAR-10 and ImageNet is marginal (less than 1% and 0.3%[SimCLR -> BYOL],respectively). And the baseline for CIFAR-10/100 is low (accuracy usually is higher than 90% on CIFAR-10 and 60% on CIFAR-100 with 800ep). If baseline is higher, then the gain may decrease.

3. SimXIR works well on fine-tuning, but there is no evidence that SimXIR can work on pretrain. My concern is that SimXIR can prevent collapsed solutions when the initialization is good (fine-tuning on pretain) but can not prevent collapsed solutions when trained with random initialization. So SimXTR may be different from non-contrastive SSL and the explanation about why non-contrastive SSL can prevent collapsed solutions may be not correctly。

4. Typos:  $L_2$-norm in Equation (3.2) should be written correctly, $|\cdot|_2^2$. $\R_C(x)$ in the next row has the same typo.

**Questions:**

See the concerns and quedstions above in the section of weaknesses.

---

> ### Author Response · Authors · 2023-11-19
> **Response to Reviewer ByEQ**
>
> We thank the reviewer for the constructive comments and insightful suggestions.
>
> **Comment 1:** Limited novelty: The method SimXIR looks very similar to Knowledge Distillation (mean teacher): SSL pretrain then knowledge distillation.
>
> **Response:** Thank you for your comment. Initially, SimXIR appears to share certain similarities with knowledge distillation, as both involve stabilizing a target network while enabling the online network to learn representations towards the target network.
>
> However, in contrast to knowledge distillation methods, our approach utilizes the online network from the previous round as the target network, deviating from the practice of maintaining an exponential average model like mean teacher. This strategic choice not only distinguishes our method but also streamlines the theoretical analysis, providing more mathematical opportunities to comprehend the inherent properties of non-contrastive SSL. This stands as a significant contribution of our work.
>
> We characterize SimXIR in the form of neighbor-averaging dynamics, shedding light on novel implicit biases such as contraction and alienation properties. These properties, previously unexplored in existing literature, contribute to a more comprehensive understanding of non-contrastive SSL.
>
> ---
>
> **Comment 2:** The gain performance on CIFAR-10 and ImageNet is marginal (less than 1\% and 0.3\%[SimCLR -> BYOL],respectively). And the baseline for CIFAR-10/100 is low (accuracy usually is higher than 90\% on CIFAR-10 and 60\% on CIFAR-100 with 800ep). If baseline is higher, then the gain may decrease.
>
> **Response:** Thank you for your insightful comment. The primary emphasis of our paper is to demonstrate that SimXIR without asymmetric structure, characterized by neighbor-averaging dynamics, can enhance representations based on a favorable initialization. Our experiments are tailored to validate this key assertion. While it is acknowledged that SimXIR shows marginal improvement for models that are well-pretrained and already exhibit strong performance, we emphasize more substantial enhancements for models with initially weaker performance, such as Relative Location and Rotation Prediction. This underscores the efficacy of SimXIR in improving representations even in the absence of asymmetric structure.
>
> ---
>
> **Comment 3:** SimXIR works well on fine-tuning, but there is no evidence that SimXIR can work on pretrain. My concern is that SimXIR can prevent collapsed solutions when the initialization is good (fine-tuning on pretrain) but can not prevent collapsed solutions when trained with random initialization. So SimXIR may be different from non-contrastive SSL and the explanation about why non-contrastive SSL can prevent collapsed solutions may be not correctly.
>
> **Response:** Thank you for your comment. As indicated by ``self-supervised fine-tuning",  SimXIR is not intended for training from scratch but tailored for fine-tuning for off-the-shelf SSL models. However, this does not mean SimXIR is different from non-contrastive SSL. In fact, its approach aligns with the principles of SSL without negative examples. SimXIR is employed for fine-tuning models with a favorable initialization, leading to substantial improvements in downstream tasks. This underscores the effectiveness of SimXIR in enhancing representation learning. Moreover, the neighbor-averaging dynamics within this framework theoretically exhibit properties of contraction and alienation, which significantly contribute to understanding non-contrastive learning. If SimXIR, without an asymmetric structure, lacked these crucial properties, the outcome would likely result in a collapsed solution.
>
> ---
>
> **Comment 4:** Typos: $L_2$-norm in Equation (3.2) should be written correctly, $|\cdot|_2^2$. $\\R_C(x)$ in the next row has the same typo.
>
> **Response:** Thank you for your careful review. We have rectified the mentioned typos and addressed some other potential errors.

---

> ### Author Response · Authors · 2023-11-23
> **Looking forward to your feedback**
>
> Dear reviewer ByEQ,
>
> Thanks again for your valuable time and insightful comments. As the deadline for the Author/Reviewer discussion is approaching, it would be nice of you to let us know whether our answers have solved your concerns so that we can better improve our work. We are looking forward to your feedback.
>
> Best regards,
>
> Authors of Paper 222

---

### Author Response · Authors · 2023-11-23
**Appreciation for Your Review and Openness to Further Comments**

Dear Reviewer,

We sincerely appreciate your dedicated time and effort in reviewing our paper. Your insightful comments have been invaluable in enhancing the quality of our work. Should you have any additional comments or queries regarding our response, we would be delighted to address them.

Thank you once again for your thorough review.

Best regards,

The Authors